# Extracellular matrix sensing via modulation of orientational order of integrins and F-actin in focal adhesions

Valeriia Grudtsyna[1,2,*] , Swathi Packirisamy[1,*], Tamara C Bidone[3], Vinay Swaminathan[1,4]

Specificity of cellular responses to distinct cues from the ECM requires precise and sensitive decoding of physical information. However, how known mechanisms of mechanosensing like force-dependent catch bonds and conformational changes in FA proteins can confer that this sensitivity is not known. Using polarization microscopy and computational modeling, we identify dynamic changes in an orientational order of FA proteins as a molecular organizational mechanism that can fine-tune cell sensitivity to the ECM. We find that $\alpha$V integrins and F-actin show precise changes in the orientational order in an ECM-mediated integrin activation-dependent manner. These changes are sensitive to ECM density and are regulated independent of myosin-II activity though contractility can enhance this sensitivity. A molecular-clutch model demonstrates that the orientational order of integrin–ECM binding coupled to directional catch bonds can capture cellular responses to changes in ECM density. This mechanism also captures decoupling of ECM density sensing from stiffness sensing thus elucidating specificity. Taken together, our results suggest relative geometric organization of FA molecules as an important molecular architectural feature and regulator of mechanotransduction.

## Introduction

Cells sense and respond to a wide range of physical cues from the ECM to regulate processes such as cell migration, proliferation, and transcription (1, 2, 3). These cues which include the composition and density of ECM proteins and associated mechanical properties such as stiffness, viscoelasticity, and architecture are converted into physical signals such as forces and deformation on the cell surface at FA sites which are then transmitted and transduced downstream via the process of mechanotransduction (4, 5, 6). The biophysical

information encoded in forces and deformations include the magnitude, its associated direction, and lastly, the rate or frequency of the physical cues (7, 8, 9, 10, 11). To decipher and decode these signals with high sensitivity and precision, cells thus need to be able to sense and separate out small changes in the magnitude, direction or loading rates along mechanotransduction pathways while maintaining fidelity of the signal intracellularly to transmit over distances to other cellular compartments and locations.

Over the years, a number of molecular mechanisms for mechanotransduction have been discovered, many of which rely on proteins undergoing force-induced changes in their function (12, 13, 14). These mechanosensory proteins or *mechanosensors*, a vast number of which localize to FA sites, can modulate their function by exhibiting catch bond behavior with interacting partners or by undergoing conformational changes because of the application of forces which results in altered protein–protein interactions or changes in their activity (15, 16, 17, 18, 19, 20). Recent studies have shown that colocalization of mechanosensors at FA sites enables physical and biochemical coupling between them and can indeed bestow sensitivity to changes in ECM stiffness (21, 22, 23). In addition, some catch-bonds in FAs have now been found to be sensitive to the direction of force acting across the interaction surface of the binding partners (24, 25). These results imply that instead of acting in isolation, mechanosensors form physical circuits along the force transmission pathway and the organization of these molecules relative to each other and relative to the forces and deformation may play a significant role in mechanotransduction. However, although we know details about the localization of these molecules and their proximity to each other, little is known about what nanoscale features determine relative organization of these molecules with each other and with the direction and magnitude of forces acting across them.

We previously showed that the primary ECM receptors, integrins can anisotropically organize and co-align with each other in FAs of cells upon activation by its ECM ligand (26). This organization which resembles the orientational order of nematic materials results in

[1]Division of Oncology, Department of Clinical Sciences, Lund University, Lund, Sweden   [2]Niels Bohr Institute, University of Copenhagen, Copenhagen, Denmark   [3]Department of Biomedical Engineering, The University of Utah, Salt Lake City, UT, USA   [4]Wallenberg Centre for Molecular Medicine, Lund University, Lund, Sweden

Correspondence: vinay.swaminathan@med.lu.se
*Valeriia Grudtsyna and Swathi Packirisamy contributed equally to this work

the orientation of integrin molecules along the long axis of FA which is also the direction of the force vector at the cell–ECM interface (27, 28). Based on these results, here we hypothesized that the relative orientation of FA proteins to each other and to the direction of force, that is, orientational order is an important feature of molecular organization in FAs and can underlie ECM sensitivity to distinct physical cues to elicit specific cellular responses. Using polarization microscopy and computational modeling, we show that indeed, integrins and F-actin are not only orientationally ordered but that this orientational ordering is altered as cues from the ECM change. For sensing of ECM density, this change in orientational order occurs in an integrin activation-dependent manner. Myosin II activity, although not required for establishing ECM density-dependent changes in the orientational order, greatly enhances it, especially at higher ECM densities, thus contributing to the sensitivity of cellular response. Finally, using a modified motor-clutch mechanism, our results show that modulation of the orientational order of FA components driven by directional catch bonds between vinculin and F-actin can capture the sensitivity of cellular responses observed while also predicting a specificity that we experimentally validate.

Taken together, these results reveal to our knowledge for the first time that in addition to nanoscale localization, the relative orientation of FA molecules with each other and relative to the forces acting across FA sites changes with ECM cues downstream of integrin activation. These changes are highly precise and can tune the downstream cellular response through mechanisms that are currently not known. In addition, our data also show that this organization and cellular response can occur through two mechanisms, one actin polymerization dependent at nascent adhesion sites, primarily at low ECM densities, and the other is myosin II-dependent which enhances overall cellular sensitivity and allows for more distinct and robust responses at higher ECM densities.

# Results

## Cellular response to changes in ECM density correlates with changes in integrin activation-dependent orientational order of αV integrins and F-actin in FAs

Having previously established the existence of orientational order of αV integrins and F-actin in FAs, here we first set out to determine if this orientational order was sensitive to changes in physical cues from the ECM. We focused on ECM density and specifically changes in density of fibronectin (FN), which binds and activates β1 and β3 integrins in our MEFs (29, 30, 31, 32).

To measure the orientational order of integrins in FAs, we used our previously developed and characterized polarization probe for αV integrin (αV-constrained[const.]) which can dimerize with β3 and β1 subunits and bind to fibronectin (26). Here, the fluorophore (GFP) is attached and constrained within the β-propeller region of the αV headpiece with a two-residue linker (26). For F-actin, Alexa Fluor (AF) 488-labelled phalloidin was used which has previously been extensively characterized for measurements of F-actin orientational order (33). We plated MEFs for 4 h on glass bottom dishes

adsorbed with varying concentrations of fibronectin (0.1, 1, and 10 μg/ml) (Fig S1A). After 4 h of plating, MEFs expressing the αV-const. probe were fixed and stained for the FA protein paxillin (for measuring integrin orientational order) or non-transfected MEFs were fixed and stained with AF 488-phalloidin and paxillin (for F-actin orientational order). Cells were then imaged using emission anisotropy total internal reflection microscopy (EA-TIRFM) as previously described (26). Here, briefly, a polarized excitation is used to illuminate fluorescent molecules in the sample and the parallel ($I_{pa}$) and perpendicular ($I_{pe}$) components of the emission are separated using a polarizing beam splitter. The relative intensities of these two components are then used to calculate fluorescence anisotropy using the formula, $r = (I_{pa} − I_{pe})/(I_{pa} + 2I_{pe})$. For calculating the orientational order of the fluorescent molecule in FAs, the average "$r$" in an FA is plotted as a function of the orientation of the FA long axis with respect to the excitation polarization, across many FAs and cells for a given condition. The amplitude (A) of the modulation in this relationship is dependent on the fraction of molecules co-aligned in-plane (XY) and out of plane (Z) and directly relates to the molecular orientational order parameter as previously determined for actin in drosophila embryos and the cell membrane (34, 35). In subsequent text and figures, this amplitude of modulation will be referred to as the orientational order.

EA-TIRFM imaging showed distinct differences between cells on different FN densities, with cells on low density showing a smaller and rounder morphology with diffused FAs compared with higher densities (Fig 1A and C). The paxillin channel was used to segment FAs and the average anisotropy "$r$" of αV integrins (Fig 1A and B) or F-actin (Fig 1C and D) was then calculated per FA and plotted as a function of the FA orientation (Fig 1B and D). For αV integrins in FAs, this modulation showed a gradual increase from an amplitude of 0.02 ± 0.005 at 0.1 μg/ml FN to 0.023 ± 0.004 at 1 μg/ml and 0.041 ± 0.008 at 10 μg/ml FN, a ~twofold increase in the orientational order (Figs 1B and S1D). For F-actin in FAs, the orientational order was ~fivefold higher compared with αV integrins at the same FN density, just as previously reported (26). On increasing FN density, F-actin showed a robust change in its orientational order from 0.11 ± 0.01 at 0.1 μg/ml FN to 0.14 ± 0.04 at 1 μg/ml and 0.2 ± 0.02 at 10 μg/ml FN, ~twofold increase (Figs 1D and S1E). To test if this change in F-actin orientational order was specifically because of changes in integrin activation which we previously found to result in orientational ordering of integrins, we pretreated cells plated on low FN density (0.1 μg/ml) with $Mn^{2+}$ which is a potent activator of integrins on the cell surface. EA-TIRFM of $Mn^{2+}$-treated cells showed the formation of distinct FAs at this ECM density compared with untreated cells and a ~twofold increase in F-actin orientational order in FAs compared with untreated cells on the same FN density (Fig S1B, C, and E). Taken together, these results suggest that changes in FN density results in a change in the orientational order of αV integrins and F-actin in FAs via specific modulation of integrin activation.

To measure local and distal to FA cellular response, we focused on the cell spread area, FA morphology, and nuclear localization of the mechanosensitive transcription factor YAP/TAZ (Fig 1E and F). Immunofluorescence staining showed that overall cell morphology on different FN concentrations on glass was clearly distinct, with cells spreading poorly with small, dispersed FAs and diffused actin

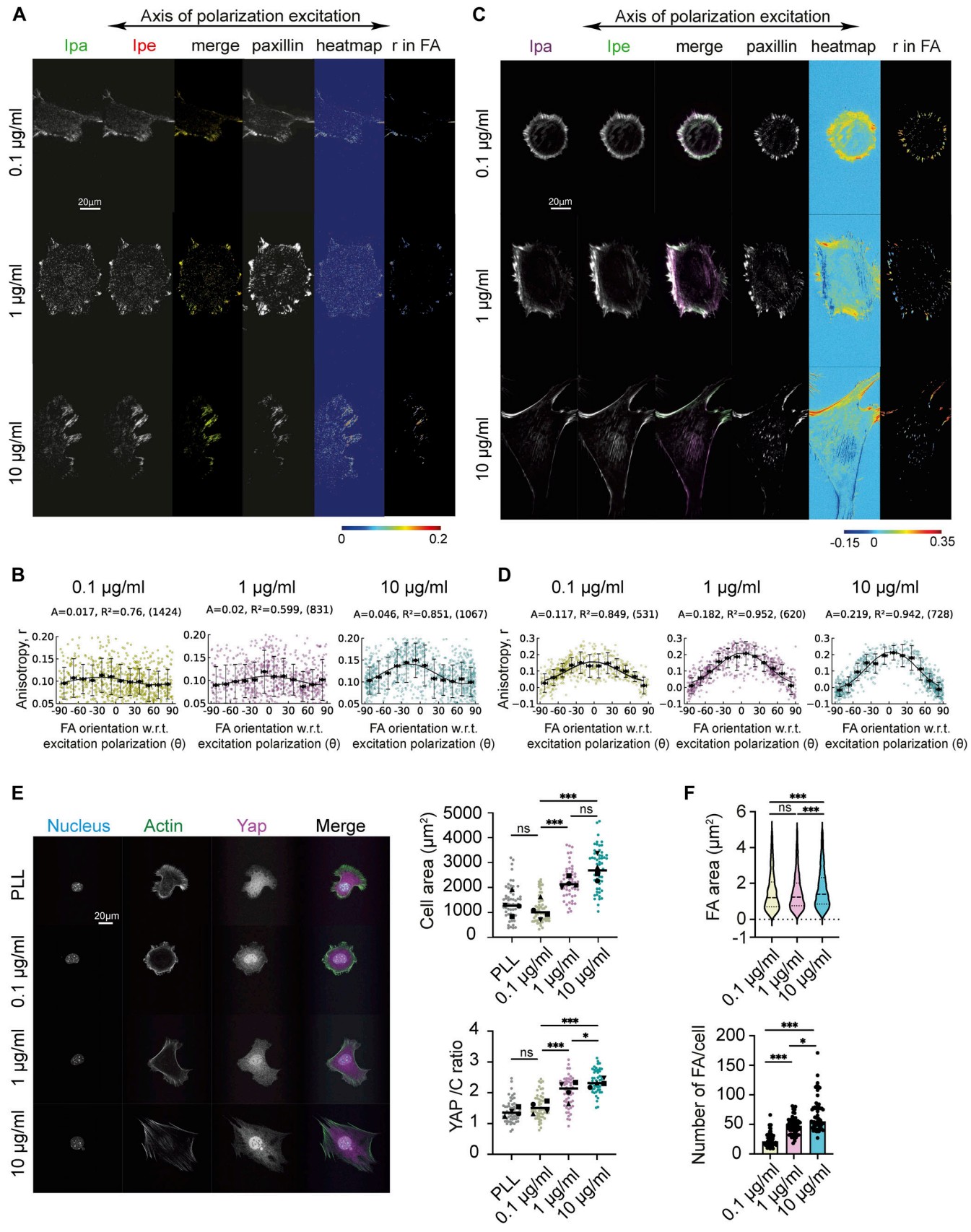

and YAP staining on the lowest fibronectin concentration and Poly-L-lysine (PLL) (Fig 1E). Consistent with previous results, we found that increase in FN density resulted in a gradual increase in the cell spread area and in nuclear to cytoplasmic YAP ratio across all three densities (Fig 1E). Locally, these FN density-dependent changes coincided with small changes in median FA size and a significant increase in the overall number of FAs formed per cell (Fig 1F). To test if these effects on cell and FA response was indeed a result of changes in FN density-induced integrin activation, we measured the cell response of cells pretreated with Mn2+ plated on low FN density and found that similar to the increase in the F-actin orientational order, there was a significant increase in cell spread area compared with untreated cells on the same FN density and an increase in the number of FAs formed per cell (Fig S1B). However, pretreatment with Mn$^{2+}$ on 0.1 $\mu$g/ml FN did not lead to a significant change in nuclear to cytoplasmic YAP ratio (Fig S1C).

Taken together, these results show that an increase in FN density leads to a concurrent increase in the orientational order of $\alpha$V integrins and F-actin in FAs in an integrin activation-dependent manner. These precise changes correlate with alterations in local FA responses including the number of FAs formed, their size, and with cell-scale responses such as cell area and nuclear translocation of YAP/TAZ.

## F-actin orientational order in nascent adhesions is highly sensitive to changes in ECM density

Within a cell, FAs are highly heterogeneous in their size, composition, and function (36). One of the key regulators of this heterogeneity is the physical coupling between FAs and the actomyosin network which results in differences in tension acting across a FA (37, 38). During its lifetime, FA undergoes myosin II-mediated changes in protein composition, size, and protein organization, all of which are critical determinants of its function (37, 39, 40, 41). This suggests that myosin II-mediated contractility is a key regulator of FA function. However, its role in FA-mediated ECM sensing is unclear as studies have shown that ECM sensing is both dependent (27, 39, 40) and independent (42) of myosin II activity. In addition, studies investigating the role of actin nucleators in the

lamellipodia have suggested an important role for actin polymerization in ECM sensing (43). Here, we wanted to investigate if ECM density-dependent changes in the orientational order and cellular response were mediated by a specific subpopulation of FAs and thus elucidate the role of myosin II and actin polymerization in this process.

As mentioned, FA size is strongly correlated with myosin II activity with small adhesions decoupled to contractility and larger FAs coupled to it. So, we re-analyzed our F-actin orientational order measurements under different conditions of FN density by binning all the FAs based on three sizes (0.1–0.25 $\mu$m$^2$; 0.25–1 $\mu$m$^2$, and 1–5 $\mu$m$^2$) (Fig 2A). Sizes smaller than 0.1 $\mu$m$^2$ precluded accurate orientation assignment and were thus not analyzed. We then plotted the average "r" as a function of FA orientation for the three different size bins and fit the modulations to extract the F-actin orientational order at each FN density (Fig 2A). This analysis first revealed that the big FAs (1–5 $\mu$m$^2$) showed ~1.45fold change in the F-actin orientational order between the lowest FN density (A = 0.17 ± 0.03 at 0.1 $\mu$g/ml) compared with the highest FN density (A = 0.24 ± 0.01, 10 $\mu$g/ml) (Fig 2A [right column], Fig 2B [largest size bin]). However, to our surprise, we also found that the biggest difference in F-actin orientational order at the different ECM densities was in the smallest adhesions (0.1–0.25 $\mu$m$^2$) with a ~1.7fold change between 0.1 $\mu$g/ml FN(A = 0.07 ± 0.01) and 1 $\mu$g/ml FN (A = 0.11 ± 0.04) and ~2.7fold difference between 0.1–10 $\mu$g/ml FN (A = 0.18 ± 0.04) (Fig 2A [left column], Fig 2B [smallest size bin]). In fact, the modulation fit on the lowest FN density was poor (R$^2$ = 0.5178 ± 0.14) leading to an unreliable measure of orientational order. Because the quality of the fit depended only on the FN density and not on the actual bin size, this poor fit at the lowest density suggests that the smallest adhesions at this density have a very disordered F-actin organization (Fig 2A). In addition, at the highest FN density (10 $\mu$g/ml FN), the difference in the orientational order of F-actin in the smallest versus the biggest FAs was relatively small (~1.4fold, Fig 2A and B [cyan line]). The middle size FA bin showed intermediate differences compared with the small and big FAs with 1.35-fold change between 0.1 $\mu$g/ml FN and 1 $\mu$g/ml FN and ~1.8fold change between 0.1 $\mu$g/ml FN and 10 $\mu$g/ml FN. Taken together, these results show that that the orientational order in the smallest FAs which coincide

**Figure 1. Cellular response to changes in ECM density correlates with changes in integrin activation–dependent orientational order of $\alpha$V integrins and F-actin in FAs.**
**(A)** Representative images of $\alpha$V-integrin-GFP transfected MEFs plated on glass coated with different FN concentrations (0.1, 1, and 10 $\mu$g/ml) imaged by EA-TIRFM. Emissions from the parallel (I$_{pa}$) and perpendicular (I$_{pe}$) channels and merge are shown (left I$_{pa}$ magenta, I$_{pe}$ green). Paxillin stained with Alexa Fluor 568 (middle). Emission anisotroiesy (r) of $\alpha$V-integrin–GFP in the whole cell (heatmap) and in the FAs are shown (right). **(B)** Magnitude of the anisotropy color scale (bottom) (B) mean integrin anisotropy (r) in FAs versus FA orientation fit to the cos$^2$ function r = C + A·cos$^2$($\gamma$ + $\theta_d$) for cells plated on each FN condition. Error bars represent SD. Each point represents one focal adhesion. "A" is the amplitude of the curve, "R$^{2}$" is the R$^2$ value of the fit, and the number of FAs is given in parenthesis. No. of focal adhesions = 712, 831, 1,067 from 40, 35, 26 cells for 0.1, 1, and 10 $\mu$g/ml conditions, respectively, from a single experiment. Plot shown for one experiment. **(C)** Representative images of MEFs on glass coated with different FN concentrations (0.1, 1, and 10 $\mu$g/ml). Cells were fixed and stained with Alexa Fluor 488 Phalloidin and imaged with EA-TIRF. Emission from the parallel (I$_{pa}$) channel, perpendicular (I$_{pe}$) channel, and a merge (left: I$_{pa}$ magenta, I$_{pe}$ green) are shown. Paxillin stained with Alexa Fluor 568 (middle). Emission anisotropy (r) of F-actin in the whole cell (heatmap) and emission anisotropy of F-actin in segmented FAs (right). **(D)** Magnitude of anisotropy color scale (bottom) (D) mean F-actin anisotropy (r) in FAs versus FA orientation fit to the cos$^2$ function r = C + A cos$^2$($\gamma$ + $\theta_d$) for cells plated on each FN condition. Error bars represent SD. Each point represents one focal adhesion. "A" is the amplitude of the curve, "R$^{2}$" is the R$^2$ value of the fit, and the number of FAs is given in parenthesis. No. of focal adhesions = 531, 620, 728 from 12 cells each for 0.1, 1, and 10 $\mu$g/ml conditions, respectively, from a single experiment. Plot shown for one experiment. **(E)** (left) Representative images of MEFs on glass coated with different FN concentrations (0.1, 1, and 10 $\mu$g/ml) and PLL, fixed, and stained with Hoechst to label nucleus, Phalloidin 488 to label actin, and Alexa 568 to label YAP. (right) Box plot quantification of cell area (top) and nuclear/cytoplasmic ratio of YAP (bottom) from analysis of immunofluorescence images of cells plated on glass coated with different FN concentrations (0.1, 1, and 10 $\mu$g/ml) and PLL. N = 57, 54, 52, 58 for each condition for cell area and N = 57, 55, 52, 59 for each condition for the nuclear/cytoplasmic ratio of YAP. **(F)** Violin plots of FA area (top) and box plot of the number of FA per cell (top) in MEFs plated on glass coated with different FN concentrations (0.1, 1, and 10 $\mu$g/ml). Quantifications were from analysis of immunofluorescence images of paxillin. No. of focal adhesions = 1,332, 2,582, 3,800 from 55, 54, 57 cells for each condition. ***$P$ < 0.001, *$P$ < 0.05, ns, not significant, Kruskal–Wallis test.

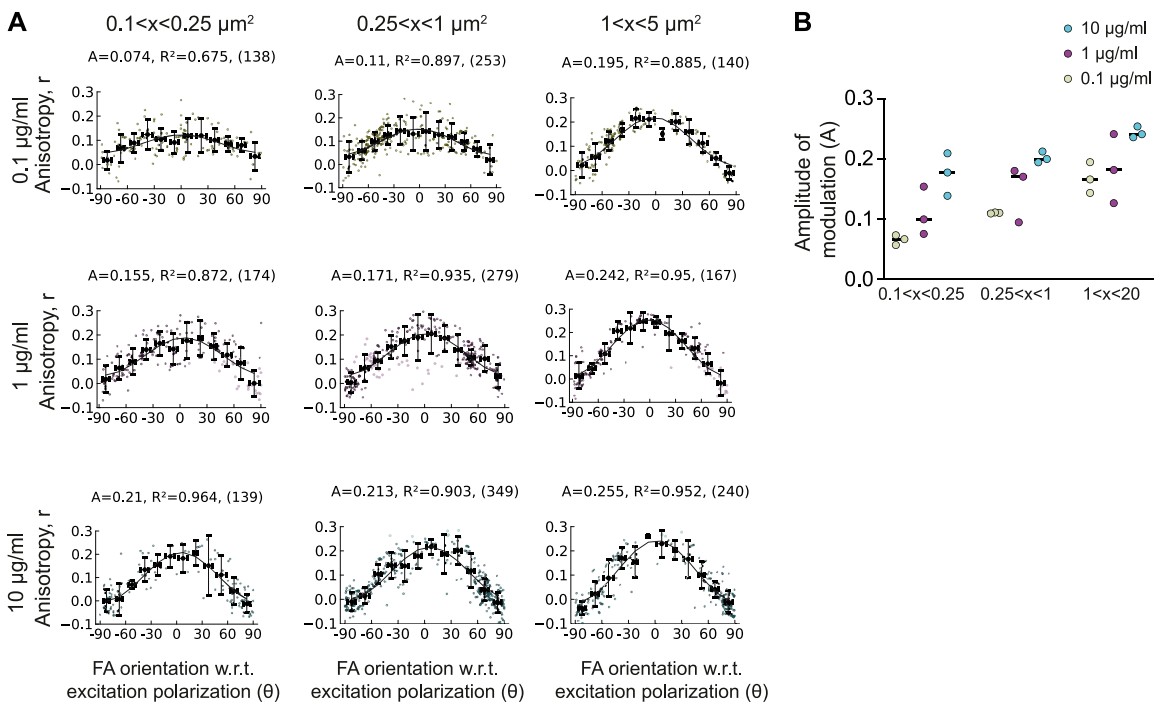

**Figure 2. F-actin orientational order in nascent adhesions is highly sensitive to changes in ECM density.**
**(A)** Mean F-actin anisotropy (r) in FAs versus FA orientation fit to the $cos^2$ function $r = C + A \cos^2 (\gamma + \theta_d)$ binned by FA size (left column) 0.1–0.25 $\mu m^2$; (middle column) 0.25–1 $\mu m^2$ and (right column) 1–5 $\mu m^2$ in cells plated on glass coated with different FN concentrations (0.1 $\mu g/ml$[top row], 1 $\mu g/ml$[middle row], 10 $\mu g/ml$ [bottom row]). Amplitude and fit values for each condition given on top. "A" is the amplitude of the curve, "$R^2$" is the $R^2$ value of the fit, and the number of FAs is given in parenthesis. **(B)** No. of focal adhesions = 138–349 from 12 cells for each condition (B) Graph showing amplitudes of F-actin anisotropy (r) binned by FA size from three separate experiments. Each experiment contains 66–349 FAs from 10–16 cells.

in size with nascent adhesions (NAs) are the most sensitive to changes in ECM ligand density and that on the highest density, F-actin exhibits the orientational order to a similar extent at all FA sizes.

### Myosin II activity is not required to establish F-actin orientational order but enhances ECM density-dependent changes in order and cellular sensitivity

Because FA size correlates with FA maturation state and myosin II activity, our results suggested that there are two distinct mechanisms for FN density-dependent changes in F-actin orientational order, one myosin II independent (in small adhesions), and one myosin II dependent (in large adhesions). To test for this, we plated MEFs on the three FN densities and treated cells with the myosin II-inhibitor blebbistatin. Inhibition of myosin II results in loss of all stress fibers and big FAs and leaves behind small NAs. However, because NAs are often harder to accurately segment and assign an orientation for measurement of orientational order, we washed out the blebbistatin for 5 min and then fixed and stained the cells with AF-488 phalloidin and paxillin to analyze the F-actin orientational order. EA-TIRFM of F-actin and analysis on the remaining small FAs after blebbistatin washout revealed that myosin II inhibition resulted in an overall lower F-actin orientational order at all FN densities compared with the untreated cells from Figs 1 and 3A. This suggests that myosin II is required for the enhanced orientational order of F-actin in all FAs. However, loss of myosin II activity did not

completely abrogate FN density-dependent changes in the orientational order of F-actin as there was ~1.18fold increase in F-actin orientational order between the 0.1 and 1 $\mu g/ml$ and a ~1.4-fold change between 0.1–10 $\mu g/ml$ FN (Figs 3A and S2). This is consistent with our previous finding that the orientational order of αV integrins can be established in the lamellipodia independent of myosin II activity (26). To test the effect of myosin II inhibition on cellular response to changes in FN density, we plated MEFs on an FN-coated cover glass at the three densities and then pretreated the cells with blebbistatin or with DMSO (Fig 3B) to measure the cell spread area and nuclear localization of YAP. Quantification and analysis showed an insignificant change in the cell spread area in blebbistatin-treated cells compared with the DMSO control over each concentration of FN (Fig 3B). This led to overall similar sensitivity in cell area across the different ECM ligand densities. Myosin II inhibition also led to decrease in YAP nuclear translocation at 0.1 $\mu g/ml$ FN and 10 $\mu g/ml$ FN, whereas there was no significant change at 1 $\mu g/ml$ FN compared with 0.1 $\mu g/ml$ FN (Fig 3B). This led to a slight decrease in ECM ligand sensitivity in overall YAP nuclear translocation with a 1.26-fold change from the lowest to the highest density (Fig 3B).

Taken together, the above results suggest that F-actin orientational order can be established independent of myosin II activity and this correlates with the ability of cells to respond to changes to FN density. However, myosin II contractility enhances this orientational order across all FN densities resulting in increased sensitivity of cells to sense and respond to changing FN density.

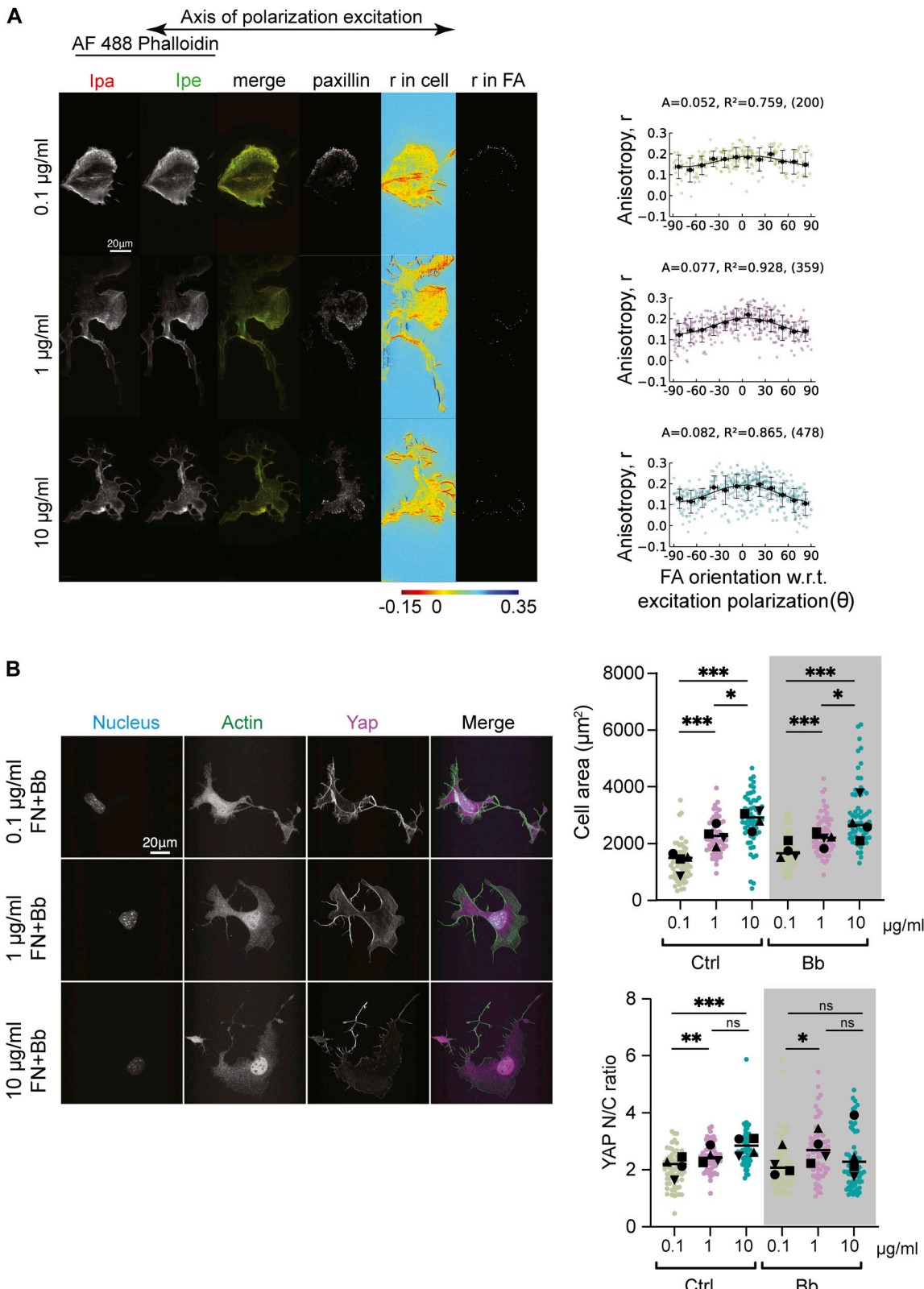

**Figure 3. Myosin II activity is not required to establish F-actin orientational order but enhances ECM density dependent changes in order and cellular sensitivity.**
**(A)** Representative images of MEFs on glass coated with different FN concentrations (0.1, 1, and 10 µg/ml) after blebbistatin treatment and a 5-min washout. Cells were fixed and stained with phalloidin 488 and imaged with EA-TIRFM. Emission from the parallel ($I_{pa}$) channel, perpendicular ($I_{pe}$) channel are shown (left). Paxillin stained with Alexa 568 (middle right). Emission anisotropy (r) of actin stained with phalloidin 488 in the whole cell (heatmap, right). Magnitude of the anisotropy color scale

## Orientational order of FA components increases sensitivity for ECM ligand binding by modulating directional catch bonds downstream of integrins

We next evaluated what mechanisms underlie the changes in the orientational order of F-actin to enable sensitivity to ECM density independent of myosin II activity. To do so, we extended our computational model of nascent adhesion assembly (42, 44), by incorporating ECM density-dependent activation of integrins and orientation-dependent kinetics of vinculin and evaluated the resulting number and lifetime of clutches in the different conditions that were tested experimentally. In the model, integrins were explicit particles that diffused on a 2D surface mimicking the ventral surface of a cell and bound explicit ligands on a bottom surface mimicking the substrate (Fig 4A). Each clutch in the model was considered formed when integrin was bound to a ligand. Two implicit components were also included in each clutch: actin with a given orientation and retrograde flow velocity, and vinculin, with a specific pathway for unbinding actin, which depended on actin orientation. During the simulations, each integrin underwent cycles of diffusion on the cell surface, binding, and unbinding of substrate ligands, and these dynamics emerged from the ECM-dependent activation of integrin, and orientation-dependent unbinding of vinculin in the clutches, which regulated the probabilities of binding and unbinding ligands, respectively. When integrin was bound to a ligand, a clutch was considered formed and the contributions from actin orientation and flow, and from vinculin kinetics were considered to determine disassembly of the clutch.

Because our experiments showed that $\alpha$V integrins' orientational order increases with increase in ECM density (Figs 1A and S1D), we incorporated a probability of integrin activation ($P_a$) that varies linearly with ECM density, with a minimum of 0.5 at 100 ligands/$\mu$m$^2$ and a maximum of 1 at 200 ligands/$\mu$m$^2$ (Fig 4C). When the integrin was active, they could bind a free ligand and form a clutch; when the integrin was inactive, it could only diffuse. When integrin was bound to a ligand, each clutch was also considered bound to the actin cytoskeleton through binding of vinculin to F-actin (via talin). Although the mechanism by which the orientational order of F-actin is established in FAs downstream of integrin activation and its orientational ordering is unknown, a previous study has shown that a long-range F-actin orientational order can be established by formation of directional catch bonds between vinculin and F-actin (25). Having explicitly measured the orientational order of F-actin here (Fig 1B), we incorporated in the model changes in F-actin orientational order by using two pathways for vinculin unbinding of actin: directional, $k_{off,+}$, and nondirectional, $k_{off,-}$ (Fig 4A and B). The directional pathway corresponded to the one with the highest peak in bond lifetime for the vinculin–actin bond at ~13 s (Fig 4B), in which vinculin forms a bond with F-actin pulling towards its pointed end. The nondirectional pathway corresponded to the lifetime–force relation with a maximum of ~3 s (Fig 4B).

To include changes in the orientational order of F-actin, we tuned the probability of vinculin unbinding actin according to the directional unbinding pathway ($P_{v,+}$) and then tested its importance in mediating ligand sensitivity (Fig 4C). To evaluate how different ECM-dependent integrin anisotropies could affect adhesion assembly, we tuned the degree by which $P_a$ varies with ligand concentration (Fig 4D) and evaluated its effect on the fraction of clutches. Lastly, to understand how the combined effects providing the orientational order of FA clutches, ECM-dependent integrin activation, and F-actin orientation affect sensing of ECM density, we tested how simultaneously varying $P_+$, and $P_a$, affected the average number of clutches (Fig 4D–F). Varying the maximum $P_{v,+}$ ($P_{v,+}$ corresponding to $n$ = 200 ligands/$\mu$m$^2$) from 0.5–1 (Fig 4C), whereas maintaining $P_a$ = 0.5 increased the average fraction of clutches from a minimum of ~0.73 to a maximum ~0.93 using 100–200 ligands/$\mu$m$^2$ (Fig 4D). Varying the maximum $P_a$ ($P_a$ corresponding to $n$ = 200 ligands/$\mu$m$^2$) from 0.5 to 1 (Fig 4C) and varying $n$ from 100–200 ligands/$\mu$m$^2$, whereas maintaining $P_{v,+}$ = 0.5, decreased the minimum fraction of clutches to ~0.71 and increased their maximum to ~0.95 (Fig 4E). Simultaneously varying $P_a$ and $P_{v,+}$ from 0.5–1 (Fig 4C) in the same range of ligand densities resulted in variations of the fraction of clutches from 0.73, up to ~0.97 (Fig 4F). The fraction of clutches increased about 1.6% using $n$ = 200 ligands/$\mu$m$^2$ and varying $P_{v,+}$ from 0.5–1, whereas keeping $P_a$ = 0.5 (Fig 4G). The fraction of clutches increased by about 4% using $n$ = 200 ligands/$\mu$m$^2$ and varying $P_a$ from 0.5 to 1, whereas keeping $P_{v,+}$ = 0.5 (Fig 4G). By simultaneously increasing both $P_a$ and $P_{v,+}$, this increase in the fraction of clutches was more than 5%. The percentage of short-lived clutches was lower using $P_a$ = $P_{v,+}$ = 1 than using either $P_a$ = 0.5 or $P_{v,+}$ = 0.5 (Fig S3A and B), meaning that clutches were ligated for a longer time when orientational ordering of both FA components (integrin and F-actin) was maximized.

Collectively, these results demonstrate that ECM ligand sensitivity increases the most by increasing both in the probabilities of integrin activation and directional catch bond. By contrast, by increasing either parameter in isolation narrows the range of the fraction of clutches, resulting in less sensitivity to ECM density. Therefore, the results from the model demonstrate that the magnitude of the orientational order of F-actin determines the precise sensitivity to ECM density through FA unbinding kinetics in an integrin-activation and orientational ordering-dependent manner.

(bottom). Mean actin anisotropy (r) in FAs versus FA orientation fit to the cos$^2$ function for cells plated on glass coated with different FN concentrations (0.1, 1, and 10 µg/ml) after a 5-min washout of blebbistatin (extreme right). Each point represents one focal adhesion. "A" is the amplitude of the curve, "R$^2$" is the R$^2$ value of the fit, and the number of FAs is given in parenthesis. **(B)** No. of focal adhesions = 200, 359, 478 from 15 cells for 0.1, 1, and 10 µg/ml conditions, respectively, from a single experiment (B) (left) Representative image of MEFs on glass coated with different FN concentrations (0.1, 1, and 10 µg/ml) after blebbistatin treatment, fixed and stained with Hoechst to label nucleus, phalloidin 488 to label actin and Alexa 568 to label YAP (right). Box plot quantification of cell area (top) and nuclear/cytoplasmic ratio of YAP (bottom) from analysis of immunofluorescence images of cells plated on glass coated with different FN concentrations (0.1, 1, and 10 µg/ml) after blebbistatin treatment. Cell area was obtained by segmenting the actin channel. N = 62, 61, 58 and 63, 61, 60 cells for cell area and nuclear/cytoplasmic ratio of YAP, respectively, for each control condition (0.1, 1, and 10 µg/ml). N = 62, 63, 61 and 60, 63, 61 for cell area and nuclear/cytoplasmic ratio of YAP, respectively, for each treated condition (0.1, 1, and 10 µg/ml). ***$P$ < 0.001, **$P$ < 0.01, *$P$ < 0.05, ns, not significant. Kruskal–Wallis test.

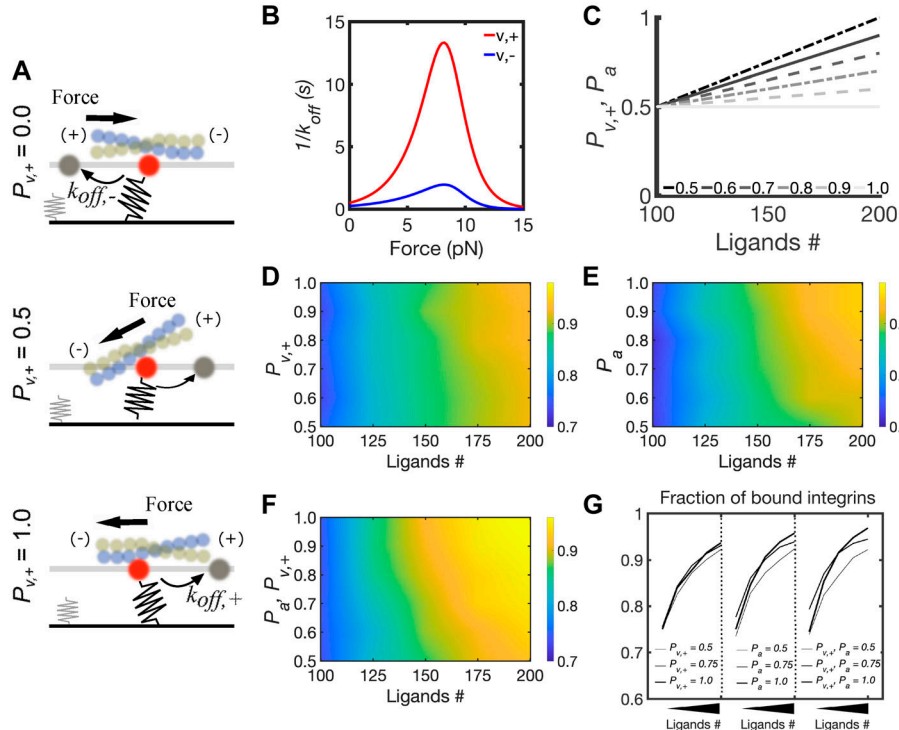

**Figure 4. Orientational order of FA components increases sensitivity for ECM ligand binding by modulating directional catch bonds downstream of integrins.**

**(A)** Schematic representation of the computational model of FA assembly based on the molecular clutch mechanism. Two 2D surfaces are placed 20 nm apart. The bottom surface represents the substrate with a random distribution of ligands, modeled as elastic springs with stiffness $k_{sub}$, which depends on the substrate young modulus, $Y$. The top surface mimics the ventral cell membrane, with integrins diffusing with diffusion coefficient $D$ and establishing interactions with substrate ligands. When integrins are active and bind substrate ligands, filaments exert a force on the clutches with a magnitude that depends on their retrograde velocity and myosin motor activity. The direction of this force determines the clutch unbinding rate, $k_{off,+}$ (directional) or $k_{off,-}$ (nondirectional). **(B)** Unbinding rates follow the lifetime ($1/k_{off}$) versus force relation, where the directional pathway of vinculins in the clutch is indicated with v,+, and the nondirectional pathway is indicated with v,-. **(C)** Linear relations between the probability of $k_{off,+}$ (directional pathway for unbinding), $P_{v,+}$, and integrin activation rate, $P_a$, with varying ligand density, $n$, between 100–200 ligands/$\mu m^2$. Legends indicate the value of $P_{v,+}$ or $P_a$ using $n$ = 200. The following relations are used: for $P_{v,+}$ or $P_a$ = 0.5 at $n$ = 200, $P_{v,+}$ or $P_a$ = 0.0 x $n$ +0.5; for $P_{v,+}$ or $P_a$ = 0.6 at $n$ = 200, $P_{v,+}$ or $P_a$ = 0.01 x $n$ +0.4; for $P_{v,+}$ or $P_a$ = 0.7 at $n$ = 200, $P_{v,+}$ or $P_a$ = 0.02 x $n$ +0.3; for $P_{v,+}$ or $P_a$ = 0.8 at $n$ = 200, $P_{v,+}$ or $P_a$ = 0.03 x $n$ +0.2; for $P_{v,+}$ or $P_a$ = 0.9 at $n$ = 200, $P_{v,+}$ or $P_a$ = 0.04 x $n$ +0.1; for $P_{v,+}$ or $P_a$ = 1 at $n$ = 200, $P_{v,+}$ or $P_a$ = 0.05 x $n$. **(D)** Average fraction of ligated clutches varying $P_{v,+}$ between 0.5–1 and $n$ between 100–200 ligands/$\mu m^2$, whereas keeping $P_a$ = 0.5. **(E)** Average fraction of ligated clutches varying $P_a$ between 0.5–1 and $n$ between 100–200 ligands/$\mu m^2$, whereas keeping $P_{v,+}$ = 0.5. **(F)** Average fraction of ligated clutches by simultaneously varying $P_{v,+}$ and $P_a$ between 0.5–1 and varying $n$ between 100–200 ligands/$\mu m^2$. **(G)** Average fraction of ligand-bound integrins for different values of $P_{v,+}$, $P_a$ or both, varying $n$ between 100–200 ligands/$\mu m^2$. All data are obtained as averages from 300 s of simulations for each condition, using $Y$ = 6 KPa.

To test the validity of our model, we next examined the implication of our experimental results on the role of myosin II on orientational order which suggests that ECM density sensing can occur in the absence of contractility. Because myosin II contractility is coupled to ECM stiffness (39), we hypothesized that FN density sensing was distinct and independent of ECM stiffness because of modulation in the orientational order of the FA components. We first tested this hypothesis in our orientational order–based motor clutch model. To do so, we systematically changed the ECM stiffness from 0.4 to 60 kPa and plotted the distribution of the fraction of clutches at three different ECM ligand densities (100, 110, and 180 ligands/$\mu m^2$) (Fig 5A). At 0.4 kPa, the fraction of clutches increased from ~0.6 to ~0.8; at 6 kPa the fraction of clutches increased from ~0.7 to ~0.95, shifting the range of ligand sensitivity upward; and at 60 kP, the fraction of clutches increased from only ~0.55 to ~0.85, shifting the range of ligand sensitivity downward. At different stiffnesses, sensitivity to ECM density also increased the most when $P_a$ and $P_{v,+}$ were increased simultaneously (Fig S3C–E). Interestingly, our model showed a biphasic response across different stiffnesses for the same ligand density (Fig 5A). This occurred because when substrate forces were too high, adhesions disassembled instead of stabilizing. Overall, the orientational order–based motor clutch model predicted ECM density sensing independent of ECM stiffness with a biphasic response across the stiffness range for a given ECM density.

We tested these predictions of our model by plating MEFs on the three different FN densities (0.1–10 $\mu g$/ml) on PA gels of the three stiffnesses used in the model (0.4, 6, and 60 KPa). We fixed and immunostained the cells after 4 h of plating for YAP, actin and nucleus, just as previously described. Quantification of the cell spread area (Fig 5B) and YAP N/C ratio (Fig 5C) across all these conditions showed a good match with the prediction from the model. At each ECM rigidity, cell area and nuclear localization of YAP increased with increasing FN density with different sensitivities. We also observed a biphasic response at the highest ECM density (10 $\mu g$/ml) across the three rigidities matching the prediction of the model.

Taken together, these results confirm that ECM-dependent changes in the orientational order of FA components and regulation of directional catch bonds provide a robust mechanism for fine-tuning the sensitivity of cells to ECM density, independent of ECM rigidity.

## Discussion

Integrin-based FAs are the primary multimolecular structures that mediate sensing of a wide range of physical cues from the ECM. At the molecular scale, FAs perform this function by assembling a network of several different proteins that are functionally and physically linked (37, 45, 46). In addition to composition, FA molecular architecture is also highly complex and thought to be a critical regulator of its function (47, 48, 49). The role of

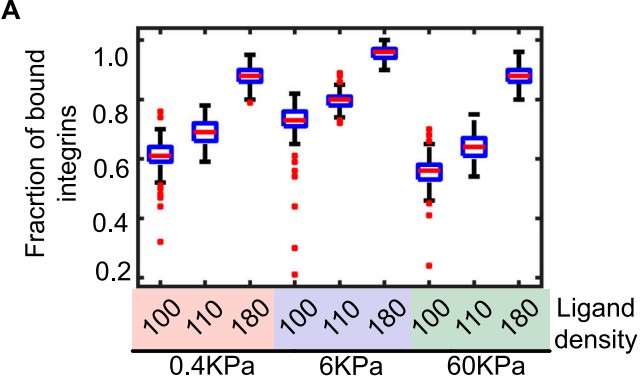

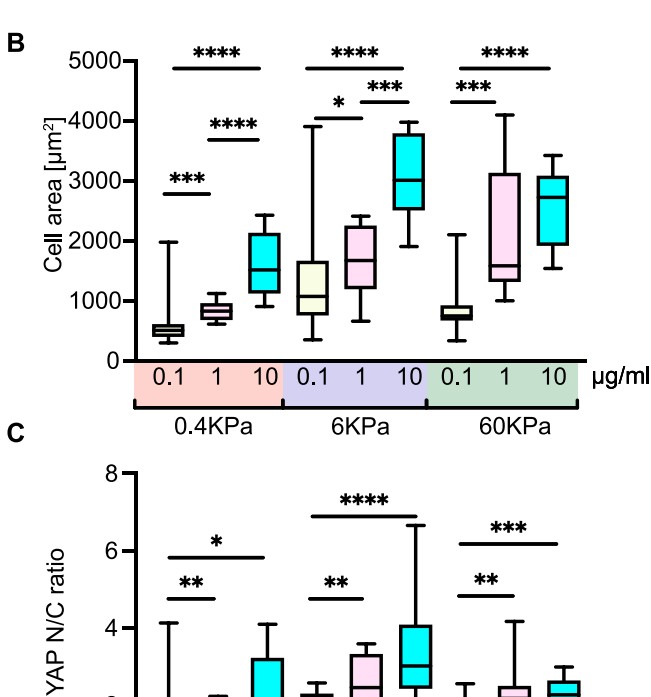

**Figure 5. Orientational ordering-dependent ECM ligand sensing predicts decoupling from stiffness of the ECM.**
**(A)** Boxplots of the fraction of ligand-bound integrins varying n between 100–180 ligands/$\mu m^2$ and using Y = 0.4, 6, and 60 kPa. All data are obtained from 300 s of simulations sampling every 1 s. **(B, C)** Box plot quantification of cell area (B) and nuclear/cytoplasmic ratio of YAP (C) from analysis of immunofluorescence images of cells plated on gels of varying Young's modulus (0.4, 6, and 60 kPa) coated with different FN concentrations (0.1, 1, and 10 µg/ml). Cell area was obtained by segmenting the actin channel. N = 12 cells for each condition.

multimolecular architecture is even more important if one considers the mechanisms of activation of mechanosensory molecules essential for FA function. The changes in protein function of mechanosensors are driven by changes in the protein structure or conformation upon application of forces which expose new binding

surfaces or cryptic sites that undergo further modifications. Owing to physical properties of force such as magnitude and direction, the geometry or orientation of interactions between forces and proteins is an important factor that can influence the changes in conformation in protein structure and thus its function (50, 51). In addition to its effect on single proteins or cluster of proteins, the relative orientational organization of different proteins, physically connected across the whole FA or FA–nucleus or FA–other cellular structures can have an impact on the nature of forces felt by distal proteins and organelles (52). However, the link between physical cues from the ECM, molecular organization within the FA and cellular response has so far been missing.

Here, we have shown that FA molecules can exhibit orientational order, that is, orientational co-alignment between similar molecules, which most importantly, changes upon small changes in physical cues from the ECM (Fig S4). Our data on F-actin and integrins in combination with our previously published data show that FAs have a molecularly anisotropic architecture that is dynamic and originates at the ECM–integrin interface and extends to the actin cytoskeleton. Mechanistically, in this study, we find two force regimes of establishment and regulation of orientational order; one which is myosin II-independent suggesting an important role for actin polymerization and forces generating by it and the other is myosin II-dependent (Fig S4). Interestingly, the role of myosin II contractility seems to be to increase the overall magnitude of orientational order of FA components and make cells more sensitive to higher ECM densities. This may correlate to conditions that require higher contractility such as the higher ECM density here or high stiffness ECMs or it may reflect a compositional switch in FAs driven by changes in integrin subtype from αVβ3 integrin to α5β1 integrin. The precise relationship between organizational (orientational order of FA components) and functional subpopulations of FAs (fibrillar versus FAs or front versus rear) will be areas of future investigations. In addition to molecular anisotropy along the plane of the membrane, it is likely that owing to the 3D nature of the FA architecture, the orientational of molecules perpendicular to the membrane (tilt) may also play a significant role in ECM sensing. Our approach here with a single probe and measurement of 2D projection of the anisotropy is currently limited in extracting the contribution of tilt, though other approaches which rely on single-molecule polarization or using linkers of different length can address this (53, 54, 55). Consistent with this 3D organization hypothesis, our previous measurements on integrin anisotropy using two different polarization probes suggest an important role for tilting of integrins upon activation (26).

Currently, the specific mechanism that results in F-actin orientational order downstream of integrin activation and orientational order is unknown. Partly, the challenge is that although the link between ECM-dependent integrin activation and its orientational order is robust, we still do not have a direct method to alter the orientational order of integrins independent of its activation. With advancements in ECM nano-deposition and nanofabrication techniques, this challenge can potentially be overcome in the future. To gain some mechanistic insight into the organizational link between F-actin and integrin orientational order, here, we use a computational model for directionally asymmetric catch bonds between vinculin and F-actin that has previously been shown to establish

long-range F-actin orientational order in migrating cells (25). Our model here shows that when the ECM-dependent modulation of orientational order measured is coupled to a directional catch bond, this can lead to highly sensitive mechanosensing of ECM cues which we also experimentally capitulate. However, it is also likely that other mechanisms contribute to the establishment and regulation of orientational order. For example, at the single-molecule level, it is possible that ECM-dependent forces acting on integrins or F-actin can directly alter the structural flexibility of the molecules resulting in changes in its orientational order. At the multimolecular scale, ECM-dependent regulation of other directional catch bonds such as between talin and F-actin or localization of actin crosslinking proteins which have previously been shown to promote F-actin orientational order in FAs could be critical for this process (24, 33, 56). Computation modeling coupled with point-mutagenesis studies in our group are currently testing the role of some of these specific interactions. Lastly, mechanistically, it is also tempting to speculate that orientational ordering of actin nucleators such as FA-specific formins can lead to F-actin orientational order in an ECM-dependent manner and this requires detailed investigation on different FA and NA-specific formins. Our EA-TIRFM images also reveal a strong correlation in the overall architecture of F-actin in the cell, away from the FAs, and the ECM-dependent changes in orientational order of F-actin in FAs. This suggests an organizational link between nanoscale orientational ordering of FA components and mesoscale F-actin organization. Interestingly, long-range mesoscale order of actin in cells has been previously shown to play a significant role in modulating ECM-dependent changes in the cell's mechanical properties (54), which mechanistically could be linked to the orientational order of F-actin in FAs.

Modulation of orientational order also offers an elegant and energetically less expensive mechanism for enhancing sensitivity to ECM cues compared with the recruitment of new proteins to the FAs and is likely applicable to other subcellular structures as well, where previously, the order of protein components has been observed (55, 57). In addition, these results further emphasis unique material properties of FAs which seem to resemble liquid-crystalline materials rather than disordered liquid-like. This may have implication in the assembly and growth of these structures in the cell and in vitro reconstitution efforts which use phase-separation methodologies (58, 59). Finally, the orientational order of load-bearing proteins has been observed across different cell types, length scales, and the animal kingdom (60, 61, 62, 63). In combination with the results of this study, this suggests a potentially conserved mechanism where the orientational order is not just the consequence of directional forces but may also be a mechanism that fine-tunes response to the forces across different cells and tissues in our body.

# Materials and Methods

### Cell culture and sample preparation

MEFs were cultured in Dulbecco's high glucose modified eagle medium (Gibco) supplemented with 10% FBS (Gibco) and 100 U/ml penicillin/streptomycin (Gibco). Polyacrylamide gels of varying Young's modulus were prepared and functionalized following a previously described protocol (64). Functionalized polyacrylamide gels or 35 mm 1.5 glass-bottom dishes (Cellvis) were coated with PLL (Sigma-Aldrich), 0.1, 1, and 10 µg/ml fibronectin (Sigma-Aldrich) for 30 min at 37°C and blocked with 2% BSA (Sigma-Aldrich) in PBS for 1 h at 37°C or at 4°C overnight. Cells were plated on FN/PLL-coated glass-bottom dishes or gels for 4 h before fixation and immuno-staining. The relative change in ECM density was verified by staining for fibronectin (Fig S1A). For blebbistatin washout experiments, cells were allowed to attach for 2 h before adding 50 µM blebbistatin or DMSO which was washed out after 2 h for 5 min. For integrin-activation experiments, cells were plated in the presence of 1 mM Mn$^{2+}$ for 4 h before fixing and staining.

### Immunostaining and transfection

Cells were fixed with 4% paraformaldehyde (Thermo Fisher Scientific) in cytoskeletal buffer (CB) (2 mM EGTA, 138 mM KCl, 3 mM MgCl, 10 mM MES, pH 6.1) for 20 min at 37°C and permeabilized with 0.5% Triton TX-100 (Sigma-Aldrich) in CB for 5 min at RT. They were then incubated with 0.1 M glycine in CB for 10 min and washed with TBS, once for 5 min and twice for 10 min. Next, the cells were blocked with 2% BSA in 0.1% tween TBS (TBST) for 1 h and immunostained with mouse anti-paxillin IgG1 antibody (610052; BD Transduction Laboratories) (1:500) overnight at 4°C. The cells were washed with TBST and incubated with secondary goat anti-mouse IgG Alexa 568 (A11031; Invitrogen) (1:400) and Alexa Fluor 488 Phalloidin (Invitrogen) (1:500) for 2 h. Samples were washed and imaged in TBS or mounted on glass slides with ProLong Glass Antifade Mount (Invitrogen).

For YAP immunostaining and cell area measurements, cells were fixed with 4% paraformaldehyde in PBS for 15 min at RT, permeabilized with 0.2% Triton TX-100 in PBS for 5 min, and blocked with 3% BSA in PBS for 45 min. Cells were immunostained with primary mouse-anti YAP monoclonal IgG2a antibody (sc-101199; Santa Cruz biotechnology) (1:400) in 1% BSA in PBS and incubated at 4°C overnight. Cells were washed with 0.05% Tween in PBS and incubated for 2 h in secondary goat anti-mouse Alexa 568-conjugated IgG antibody (Life Technologies Corporation), Alexa Fluor 488 Phalloidin (Invitrogen) (1:500) and nuclear dye Hoechst 350 (33342 Solution; Thermo Fisher Scientific) (1:4,000). Samples were washed and imaged in TBS or mounted on glass slides.

To measure anisotropy and orientational order of αV Integrin, αV Integrin–GFP-constrained plasmid based on previous work was used (26). Cells were co-transfected using the Neon Transfection system (MPK5000; Invitrogen) with 2,500 ng of αV Integrin–GFP-constrained plasmid and untagged β3 integrin plasmid at 1,500 V, 30 ms, 1 pulse. Transfected cells were cultured in antibiotic-free media for 48 h before plating on FN/PLL-coated 35-mm 1.5 glass-bottom dishes for 4 h. The cells were fixed and stained for paxillin.

### Emission anisotropy-TIRFM for measurement of orientational order

Images were acquired using total internal reflection fluorescence microscopy (TIRF) configuration on a Nikon Eclipse Ti microscope with the following available laser lines: 405, 488, 561, and 657 nm

and Spectra EX (Lumencor). TIRF APO 100x 1.49 N.A. objective was used for acquiring images. Emission/excitation filters used were as follows: GFP (mirror: 498–537 nm and 565–624 nm; excitation: 450–490 nm and 545–555 nm; emission: peak 525 nm, range 30 nm) and mCherry (mirror: 430–470, 501–539, and 567–627 nm; excitation: 395–415, 475–495, and 540–560 nm; emission: peak 605 nm, range 15 nm) or Continuous STORM (mirror: 420–481, 497–553, 575–628, and 667–792 nm; excitation: 387–417, 483–494, 557–570, and 636–661 nm; emission: 422–478, 502–549, 581–625, and 674–786 nm).

A polarized evanescent wave excites the fluorophores. In the emission pathway, the emitted light was split into p and s polarization components with a polarization beam splitter (Laser Beamsplitter zt 561 sprdc), placed into an Optosplit III(Cairn Research). The orthogonal images (Ipa and Ipe) were projected onto two separate fields of view, manually aligned, and captured with a Teledyne Photometrics 95B 22 mm camera. The polarization of the evanescent field was verified by measuring the extinction coefficient.

### Confocal and widefield microscopy, calculation of YAP N/C ratio and FA analysis

Images were acquired using a Nikon Confocal A1RHD microscope, with 405-, 488-, 561-, and 640-nm laser lines and equipped with GaAsP PMTs or a Nikon Ti2-E widefield fluorescence microscope, with 405, 488, and 561 nm LED light source (Lumencor SpectraX light engine). The following objectives were used for imaging: TIRF APO 100x oil 1.49 NA, Plan Apochromal Λ 60x 1.42 NA and Plan APO Λ 100x oil 1.45 NA. To capture the entire cell volume, z-stacks with a 0.6-$\mu$m step size for cells on glass and a 1-$\mu$m step size for cells on gels were taken (10–20 steps per cell).

Single-plane images and maximum projections of the z-stack in YAP channel were background-subtracted using functions "sigma_clip" and "estimate_background" provided by the Photometry.jl. Median filtering was applied to images taken with 100x objective. The nucleus was segmented from the summed-up nuclear channel z-stack and eroded (with five pixels for the images taken with 60x objective and seven pixels for images taken with 100x objective) to represent the area with nuclear YAP. The nuclear segment was dilated by 55 (for 60x) or 59 (for 100x) pixels and subtracted with a nuclear segment dilated by 10 (for 60x) or 14 (for 100x) pixels to attain a ring-shaped segment around the nucleus. This way, the transition area between the nuclear and the cytoplasmic region was excluded. The cell was segmented in the actin channel to represent the cell spread area. The intersection of the ring-shaped segment and the slightly eroded actin segment represented the area with cytoplasmic YAP. For computation of N/C YAP ratio, the median YAP intensity was divided by the median cytoplasmic YAP intensity. The entire cytoplasmic area was not used to avoid the contribution of low YAP counts caused by thin cell edges. The Julia code is available on request.

Paxillin images, taken as previously described in TIRF mode, were submitted for analysis to the online FA Analysis Server (65). The adhesion size options were set to 0.10 μm² < FA < 5 μm².

### Calculation of orientational order

All image analysis for anisotropy were performed using Julia (v.1.6.0) based on a previously described analysis pipeline (26). The codes are available on request.

The Ipa and Ipe images were aligned using QuadDIRECT (https://github.com/timholy/QuadDIRECT.jl). G-factor, $G = \frac{Ipa}{Ipe}$ was calculated in conjunction with every imaging session to correct for polarization bias in the detection system and the optical path. A low-concentration solution of fluorescein (Sigma-Aldrich) in water was imaged with the same camera settings, using a 488-nm LED for excitation (66). The resulting Ipa and Ipe images were used for G-factor computation.

The fluorescence anisotropy formula,

$$r = \frac{Ipa - G.Ipe}{Ipa + 2.G.Ipe},$$

was applied to the F-actin-phalloidin or const. $\alpha$V integrin images to create an anisotropy heatmap. FAs were segmented from the background subtracted paxillin image using the "Moments" binarization algorithm from the ImageBinarization.jl package. The FA mean anisotropy and the angle between the FA long axis and the excitation polarization ($\theta$) were extracted using in-house packages in Julia. FAs between 0.1–5 $\mu m^2$ and with eccentricity less than 0.9 (and less than 0.7 in the Bb washout experiments) were filtered out for reliable estimation of FA long axis orientation.

Mean anisotropy values (mean intensities from the anisotropy heatmap) from all FAs, collected from all cell images within the same condition, were plotted against the corresponding $\theta$ angles. The data were binned into 15° bins (12 bins in total) and the mean "r" together with SD was calculated. The following trigonometric function was fitted to the binned means using the "curve_fit" function from the LsqFit.jl package:

$$r = A.cos^2(\theta + p) + C,$$

where r is the fluorescence emission anisotropy; amplitude A is directly related to the orientational order; $\theta$ is the angle between the FA long axis and the excitation polarization; p is the phase shift, which defines the average orientation of dipoles in the ensemble of fluorophores; C is the anisotropy offset because of the background. Negative counterclockwise angles were converted to positive angles.

### Molecular clutch model of adhesion assembly

To understand how the orientational order of actin filaments could enable ligand sensitivity, we extended our computational model of nascent adhesion assembly at the leading cell edge. Like previous approaches from us and others (42, 67, 68, 69), our model was based on the molecular-clutch mechanisms, in which adhesion clutches intermittently transmit cytoskeletal force to the substrate and substrate rigidity to the actin flow. Cytoskeletal force was produced by intracellular myosin motors regulating the actin flow velocity and the number of engaged clutches. Substrate force regulated the strength of the integrin-ligand binding, mimicking integrin-fibronectin bonds in terms of lifetime versus force relations. Cytoskeletal force regulated the strength of the integrin–actin binding, mimicking vinculin–actin bonds in terms of directional versus nondirectional lifetime versus force.

Each clutch in the model was represented explicitly in 3D, with an integrin particle bound to a ligand particle; the force across the clutch and its lifetime were calculated by implicitly considering the

contributions of actin flow velocity and orientation, and vinculin kinetics in response to actin orientation. Once an integrin bound a substrate ligand, it was considered also bound to the actin cytoskeleton through vinculin. The lifetime of the fibronectin and actin bonds depended on the kinetics of the bonds of integrin with substrate ligand and of vinculins with actin. It has been shown that both integrin–fibronectin bond and vinculin–actin bonds are catch–slip bonds, meaning that their lifetime first increases as a function of force, then decreases as the force increases further ([16], [25]).

Initially, a given number of fibronectin molecules were randomly distributed on a substrate, to which integrins could bind. The substrate was represented as an isotropic and elastic material, consisting of a several ideal springs which provided binding sites for integrins.

Integrins underwent cycles of diffusion along the top surface of the model mimicking the ventral membrane of cells, followed by activation, binding, and unbinding of substrate ligands underneath it (Fig 3A). As active integrins bound substrate ligands, they formed a clutch and experienced cytoskeletal force, from vinculins connections with actin undergoing retrograde flow, as was seen in the lamellipodium ([70]). The number of vinculin–actin bonds depended on force and varied between two and 11 ([71]). When either integrin or at least one vinculin were bound, the clutch was considered engaged. When the clutch was engaged, it transmitted the actin flow to the substrate. This tension was used to determine the unbinding rates of vinculins and integrin. Therefore, these clutches governed local balances between cytoskeletal contractile force and substrate stiffness. When both integrin and all vinculins in a clutch were unbound, the clutch was disengaged and integrin became free to diffuse, mimicking the free diffusion of unligated integrin receptors on the ventral surface of cells ([72]). The actin flow was modulated by the number of clutches engaged with the substrate and the number of molecular motors in the system. The force on the integrin–fibronectin bonds was also directly proportional to substrate stiffness. All parameters in the model were based upon available experimental data and previous modeling approaches ([73]). We directly incorporated the lifetime versus force relationships of integrin–fibronectin and vinculin–actin bonds from atomic force microscopy and optical trap single-molecule experiments ([16], [25]). To account for ligand-dependent integrin activation, we varied the probability of integrin activation, $P_a$, proportionally with ligand density To account for different degrees of actin filaments orientation, we systematically varied the probabilities of directional catch bond for vinculin, $P_{v,+}$. (Fig 3C). To quantify the number of clutches, we measured the average fraction of ligated integrins from simulations of 300 s. More details about the model implementation and parameters are reported in the additional notes below.

### Additional notes on computational model

We used a Brownian Dynamics approach to simulate the formation and disassembly of adhesion clutches in response to actin filament orientations, ligand density, and substrate stiffness. The model considered integrins, ligands, vinculins, and actin filaments. Integrins and ligands were represented as explicit point particles in the simulation domain, whereas actin filaments and vinculins were considered implicitly, using kinetic parameters accounting for orientational order.

### Integrin activation rate indicates integrin directionality

Integrins switch between inactive and active states, and only when active do they form a bond with free substrate ligands. Our current and previous experimental measurement directly links ECM-dependent integrin activation with the orientational order of integrins in FAs. The model therefore couples the probability of activation, $P_a$, with the change in orientational order measured in the experiments.

### Directional vinculin catch bond kinetics indicates actin orientation

In the model, actin filament, actin flow, and vinculin proteins are modeled implicitly through $v_{flow}$, and $k_{off}$. The flow velocity, $v_{flow}$, displaces ligated clutches to build tension on their bonds with substrate ligands. Depending on the magnitude of this tension, a number of vinculin is considered (two for forces below 8 pN; five for forces between eight and 15 pN; nine for forces between 15 and 21 pN; 11 for forces > 21 pN). The pathway for vinculin unbinding (which determines the unbinding rates given certain force values) is chosen, in each run, depending on the probability of directionality that the model is testing. When it is assumed that actin filaments are directional (barbed end pulled), the first pathway (with maximum bond lifetime equal to 13 s) is used. Vice versa, when actin filaments are not directional (pointed end pulled), the second unbinding pathway (with maximum bond lifetime equal to 3 s) is used for all vinculins. For intermediate probabilities of directional versus nondirectional actin filaments, then a proportional number of vinculins will unbind with one pathway and the remaining will follow the second pathway.

### Simulation domain and boundary conditions

Like our previous implementation of the adhesion assembly model ([42], [44]), the computational domain was 3D and consisted of two parallel surfaces of 1 $\mu m$ per side, separated in the vertical direction by $L$ = 20 nm, a dimension typical of the extension of the integrin headpiece ([74]). The bottom surface represented the substrate with immobilized point particles mimicking ligands; the top surface represented the ventral membrane of a cell, with integrins crossing it to form connections both intracellularly and extracellularly (Fig 3B). Integrins were initially randomly distributed on the top surface and diffused with coefficient $D$ = 0.29 $\mu m^2/s$ ([72]) until they established a connection with a substrate ligand. Once a connection was formed, a clutch was considered formed. In the clutch, integrins were also considered bound to actin through talin and vinculin. The force from actin caused vinculin to unbind actin and integrins to unbind ligands. Once all connections were lost in the clutch, the clutch became disengaged and integrins unbound their ligands to restart diffusing. To avoid finite size effects on the diffusive motion of integrins, periodic boundary conditions in the lateral directions of the domain were applied.

## Integrin and ligand representation

Each $i$-th integrin and $j$-th ligand was defined by 3D position vectors, $r_i$ and $r_j$, respectively. The vector $r_i$ presented $x$, $y$, and $z$ coordinates of the $i$-th integrin; the vector $r_j$ presented $x$, $y$ and $z$ coordinates of the $j$-th ligand. At every timestep of the simulations, $x$ and $y$ of $r_i$ were updated to track integrin displacement, whereas $z$ remained fixed at $z = 0$ $\mu m$; $x$, $y$, and $z$ of $r_j$ remained all fixed over the course of the simulations as ligands were immobilized, with $z = -0.02$ $\mu m$.

## Clutch representation

Diffusive integrins could activate with probability $P_a$ and once active, they were able to bind free ligands. Specifically, when an active integrin came in proximity of a free ligand (<21 nm from it), it bound the ligand by establishing a harmonic interaction. This interaction presented spring constant $k_{sub}$ (proportional to the substrate Youngs' modulus, $Y$), and equilibrium distance $L$ ([74]). Once a connection between integrin and ligand was formed, the clutch was considered formed. As a clutch, integrins could also interact with actin. Actin flow velocity acted on each ligated integrin as a force parallel to the substrate, resulting in the buildup of tension across the clutch. Depending on the magnitude of this tension, a certain number of vinculin-actin bonds were assigned to the clutch. For forces below 8 pN, two vinculin–actin bonds were considered; for forces between 8 and 15 pN, five vinculin–actin bonds were considered; for forces between 15 and 21 pN, nine vinculin–actin bonds were considered; for forces > 21 pN, 11 vinculin–actin bonds were considered ([71]). The model assumed that, before integrin could restart diffusion, not only the bond that the integrin itself formed with the ligand needed to break but also all bonds that vinculin had formed with actin needed to fail. Bond breakages depended on unbinding rates, which were determined by the lifetime versus force relations. These relations followed different types of catch–slip bonds: one for integrin–ligand and two for vinculin–actin bonds. One of the vinculin–actin lifetime versus force relation corresponded to the perfect alignment of the filaments with the integrins and the other indicated perfect misalignment of the actin filaments with integrins. Catch–slip bonds presented a decreasing dissociation rate constant with increasing force first, followed by an increasing dissociation rate constant as force was increased further.

## Force acting on the clutches

The total force acting on each $i$-th clutch, $F_i$, resulted from the sum of a stochastic and a deterministic contribution. A stochastic force, $F_T$, was applied to all clutches in the membrane at every $dt$, to mimic thermal effects generating diffusion. This force satisfied the fluctuation–dissipation theorem. Deterministic forces originated from actin flow, $F_{flow}$, and substrate tension, $F_{sub}$. Thus, the total force acting on each $i$-th clutch was calculated as:

$$F_i = F_T + F_{flow} + F_{sub}$$

## Lifetime of clutch engagement depends on force

The clutch persisted for a characteristic lifetime which depended upon the force on the clutch, $F_i$. This force determined integrin–ligand unbinding rate, $k_{off,integrin}$, and vinculin–actin unbinding rates, $k_{off,+}$ or $k_{off,-}$, which depended on the orientation of actin in the model. The clutch was considered disassembled when all bonds were disassembled (all vinculin–actin bonds and the integrin–fibronectin bonds), at which point integrins restarted diffusing.

The catch–slip bonds followed the formalism originally proposed by Bell, which included a strengthening pathway followed by a weakening pathway indicated as a sum of two exponentials with opposite signs ([75]). For the integrin–ligand bond, the unbinding rate was:

$$k_{off} = 2*e^{-0.0640*F} + 0.00005*e^{0.26*F},$$

For each vinculin–actin bond, the unbinding rate also depended on filaments orientation and therefore it depended on the direction of force application. For forces applied toward the filament's pointed end, the catch bond was directional, with the longest lifetimes and corresponding unbinding rate:

$$k_{off,-} = 2*e^{-0.046*F} + 0.00005*e^{0.78*F}$$

For forces applied toward the filament barbed end, the catch bond was nondirectional, and the unbinding rate was:

$$k_{off,+} = 4*e^{-0.28*F} + 0.00005*e^{0.95*F}$$

## Substrate model

The substrate was considered an elastic solid, consisting of randomly distributed ideal linear springs, mimicking individual fibronectin molecules, providing binding sites for integrin, and with stiffness depending on substrate rigidity, as:

$$k_{sub} = \frac{YA}{L},$$

where $Y$ is the Young's modulus (we tested values in the range 0.4–60 kPa), $A$ is the integrin/ligand cross-sectional area (corresponding to 80 nm$^2$, from an ideal bar of radius ~5 nm, corresponding to approximately half the value of the transmembrane leg separation of an integrin in the open conformation), and $L$ is the equilibrium distance between integrin and bound ligand.

Hooke's law for each spring in the bundle can be written as:

$$F_{sub} = k_{sub}\Delta L,$$

where $\Delta L$ is the variation from the equilibrium separation between integrin and bound ligand. We used ligand densities between 100 and 200 ligands/$\mu m^2$.

## Actin flow

Actin was considered as a force acting on integrin which depended upon the retrograde flow velocity ($v_{flow}$). It was calculated as:

$$\boldsymbol{F_{flow}} = \zeta_i v_{flow}$$

$v_{flow}$ was calculated at each timestep of the simulations through the linear force–velocity relationship (68, 76):

$$v_{flow} = v_u \left( 1 - \frac{x_c k_{sub}}{n_m F_m} \right)$$

where $v_u$ = 0.11 $\mu$m/s is the unloaded velocity, $x_c$ is the number of bound clutches, $n_m$ = 135 is the number of motors, and $F_m$ = 2 pN is the motor stall force.

## Implementation algorithm

Recognizing that inertia is negligible on the length and time scales of integrin motion in the plasma membrane, the displacement of each i-th integrin was governed by the Langevin equation of motion in the limit of high friction (77)

$$\frac{d\boldsymbol{r_i}}{dt} = \frac{\boldsymbol{F_i}}{\zeta_i},$$

where $r_i$ was the position vector of integrin; $\zeta_i$ was integrin friction coefficient calculated using the Einstein relation, as $\zeta_i = \frac{k_B T}{D}$ = 0.0142 pN s/$\mu$m, where $k_B T$ = 4.11 pN nm and $D$ = 0.29 $\mu m^2$/s; $dt = 10^{-4}$ s was the simulation timestep; $F_i$ was the total force on the clutch, including a stochastic contribution from thermal effects and a deterministic contribution from actin flow, governed by the amount of ligated clutches, number of motors and their stall force, and substrate mechanics.

Considering all forces acting on integrins at every timestep, their positions were updated iteratively using an explicit Euler integration scheme:

$$\boldsymbol{r_i}(t + dt) = \boldsymbol{r_i}(t) + \frac{d\boldsymbol{r_i}}{dt}dt = \boldsymbol{r_i}(t) + \frac{\boldsymbol{F_i}}{\zeta_i}dt.$$

# Data Availability

All raw data that support the findings and the image analysis codes are available from the corresponding authors upon request.

# Supplementary Information

# Acknowledgements

We thank P Nordenfelt, Amin Doostmohammadi, Martijn Gloerich, Sebastian Wasserstrom, and all the members of the laboratory of cell and molecular mechanobiology (LCMM) for their discussion and support. Johannes Kumra Ahnlide is specially acknowledged for developing the initial analysis pipeline and all the help in developing code and maintaining image storage servers. Lund University Bioimaging Centre (LBIC) at Lund University is gratefully acknowledged for providing experimental resources. This research was funded by the Knut and Alice Wallenberg Foundation (V Swaminathan), Wallenberg Centre for Molecular Medicine, Lund; Cancerfonden (V Swaminathan, 19 0445 Pj, Projekt grant) and the National Science Foundation (TC Bidone, NSF BMMB 2044394).

## Author Contributions

V Grudtsyna: conceptualization, software, formal analysis, investigation, methodology, and writing—review and editing.
S Packirisamy: formal analysis, validation, investigation, and writing—review and editing.
TC Bidone: conceptualization, software, formal analysis, investigation, and writing—original draft, review, and editing.
V Swaminathan: conceptualization, resources, supervision, funding acquisition, investigation, methodology, project administration, and writing—original draft, review, and editing.

## Conflict of Interest Statement

The authors declare that they have no conflict of interest.

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
