## [Reviewer comments · Life Science Alliance]

Life Science Alliance

Extracellular matrix sensing via changes in orientational order of integrins and actin in adhesions

Valeriia Grudtsyna, Swathi Packirisamy, Tamara Bidone, and Vinay Swaminathan

DOI: <https://doi.org/10.26508/lsa.202301898>

Corresponding author(s): Vinay Swaminathan, Lund University

Review Timeline:

Submission Date:	2023-01-04
Editorial Decision:	2023-02-15
Revision Received:	2023-05-18
Editorial Decision:	2023-06-21
Revision Received:	2023-06-30
Accepted:	2023-07-03

Transaction Report:

February 15, 2023

Re: Life Science Alliance manuscript #LSA-2023-01898-T

Dr. Vinay Swaminathan
Lund University
BMC D14
Sölvegatan 19
Lund, Skåne 223 62
Sweden

Dear Dr. Swaminathan,

Thank you for submitting your manuscript entitled "Extracellular matrix sensing via changes in orientational order of integrins and actin in adhesions" to Life Science Alliance. The manuscript was assessed by expert reviewers, whose comments are appended to this letter. We invite you to submit a revised manuscript addressing the Reviewer comments.

Thank you for this interesting contribution to Life Science Alliance. We are looking forward to receiving your revised manuscript.

Sincerely,

B. MANUSCRIPT ORGANIZATION AND FORMATTING:

Reviewer #1 (Comments to the Authors (Required)):

In the manuscript by Grudtsyna, V. et al the authors apply computational and cellular biology approaches to assess the role of orientational order on cytoskeletal transmission of forces from ECM-engaged integrin adhesion receptors. This work builds upon a previous publication (Swaminathan, V. et al, 2017, PNAS) where the constrained α V integrin probe was applied to demonstrate that integrins are aligned by the forces applied by the actin cytoskeleton in conditioned with directional motion. The more recent findings in the current work support this alignment of integrins by the actin cytoskeleton, finding that increased ligand (FN) density occurs in conjunction with an increase in orientational order. Furthermore, assessment of small and larger adhesions, as well as the uncoupling of myosin II activity through inhibition with Blebbistatin, revealed the importance of myosin II activity in detection and response to ligand density.

Comments on main points:

The first main point of the paper is that orientational order in both α V and F-actin increases with ligand density, and that this is linked to integrin activation state. To support this finding,

Related to Fig. S1, can the authors comment on the increased spreading that is occurring independently of changes in focal FA area in Fig. S1c? Like in Fig. 1f, was there also a decrease/increase in FA number?

Further to Fig. S1, given the link between adhesion signalling and YAP nuclear localisation, can the authors confirm whether this increase in cell spreading from Mn^{2+} treatment also results in an increase in YAP nuclear localisation? Can the authors also provide representative images for the data in Fig. S1c to help the reader to better appreciate the significance of the differences?

In Fig. 2, the authors compare nascent and larger focal adhesions based on the relative areas. This data is quite difficult to evaluate, as the smallest bin for the 0.1 μ g/ml FN has a very poor R2 for the fit. Is it possible to increase the size of the bin to include the middle values, where the R2 is consistently higher than 0.9, by increasing the size range from 0.1-1 μ m² to improve the fit quality for the 0.1 μ g/ml.

Also, it is unclear from the figure legends the number of separate biological replicates that were conducted. Could the authors add this detail into the legends to help the reader to better evaluate the data? This also brings with it the issue of cell numbers in many experiments being very low. I can understand that for the polarisation microscopy, the numbers are suitably high as individual FAs are being assessed, but for the YAP N/C ratio assessment, as well as the cell size and FA #/cell (Fig. 1e/f and Fig. 3b), the cell numbers are far too low to draw such strong conclusions. I would hope to see at least 60 cells (20 from each of three biological replicates) for each condition before publication.

Reviewer #2 (Comments to the Authors (Required)):

The work « Extracellular matrix sensing via modulation of orientational order of integrins and F-actin in focal adhesions » by V. Grudtsyna et al. uses a method already developed by the authors, based on polarized fluorescence imaging to explore the intrinsic relation between the orientational order of these proteins and the properties of the extra cellular matrix (ECM rigidity and density). The use of this method is made possible by the measurement of integrin and actin proteins with appropriate labelling, e.g. which exhibit sufficiently reduced angular fluctuation, thanks to reduced linker length and flexibility between the fluorophore and protein of interest. Despite remaining fluctuations, the method allows in principle for quantitative results when comparing different situations involving the same protein construct. The authors perform several test experiments (including varying the properties of the ECM substrate and actomyosin contractility) and based on a model, conclude that the orientation of focal adhesion (FA) proteins participate in the regulation of mechanotransduction, which is the main result of the work. The work is of interest and the application of the method brings extra knowledge on the orientational organization of proteins, which is a need when addressing the main question of the relation between orientation and mechanotransduction. There are however points that need to be better addressed in order to support the conclusion of the work.

1. The method used in the work has the advantage of being simple, however it presents several inconvenients that make its capabilities of quantification challenging. This is not addressed in the manuscript :
 - 1.1 The method requires a lot of different orientation angles of FAs being measured to extract a reliable anisotropy (order) value from the measured proteins. What is the required number of angles measured to be able to accurately measure an anisotropy value ? finding FAs of different orientations can be difficult and require a lot of data to be recorded, how is the possible heterogeneity between FAs taken into account in the anisotropy curves results ?
 - 1.2 The experiment is an intrinsic projection of the information in 2D, since pure polarization projections are used. This means that if the disorder is slow but is tilted in 3D, this will result in a measurement that shows a high disorder, due to the well problem issue of bias introduced by 3D orientation of the measured organizations. How do the authors ensure that this is not an issue ? can some of the measurements, in particular decrease of anisotropy values, be affected by 3D tilt rather than disorder, in particular when measuring integrin (which 3D orientation varies a lot) and actin (which is tilted in the FA structure) ?
 - 1.3 The measured anisotropy value is a mixture of what is due to structural disorder and what is due to the orientational freedom fluctuation of the fluorescent molecules attached to the protein of interest. Is there a good proof that this fluctuation does not change in the different measurements performed by the authors ? could the anisotropy changes measured be due to a modification of this fluctuation behavior, due to forces exerted on proteins (which is necessarily happening in the studied process), or due to different packing properties ?
2. The number of data point in modulation measurements is not clear: it is said that there are about 1000 FAs measured from collections of 35 to 80 cells (for instance in figure 1), however this would mean about 20 to almost 40 FAs per cells, which does not seem to be the case in some of the situations shown in Fig. 1 a,c.
3. Fig. 1f shows in the label a number of FA per cell of 600, this is very surprising.
4. The measurement of the orientation of the measured FAs is not clearly explained. This orientation seems to be reported with a high precision in the anisotropy curves, while there can be a strong error bar around a given FA orientation. In small FAs in particular the shape does not allow to visualize a clear orientation, its value can be defined with variations up to 40 degrees, how is the mean orientation error accounted for ?
5. The type of actin population chosen for the measurement is not clearly defined with respect to the question addressed in this work. Shouldn't actin be investigated at the FA proximity ? in the images shown in Fig. 1 and Fig. 3, there seem to be a lot of actin populations from peripheral stress fibers, rather ventral stress fibers which involve more clearly the connection to FAs.
6. In the clutch model developed in the Supplementary Note on computational model, what seems to be directional is the direction of actin flow, while in the rest of the paper molecular orientation is driven by the probability Pv1 introduced in the model. The authors should make clearer where the orientation of actin vs. flow comes into play in the model. it is also not clear how the integrin directionality measured in the experimental section is related to directionality in the theoretical model.
7. The finding that 'Orientational order of FA components increases sensitivity for ECM ligand binding by modulating directional catch bonds downstream of integrins.' Is the result of simulations, and it is not clear how this model is related to the data measured in the rest of the work. This is an important conclusion since the main point of the manuscript is to address if FA molecules orientation is the result or the cause of the mechanosensitive process leading to integrin activation.
8. The authors could probably consider in the discussion part of their work, how the question addressed in the previous point can be addressed experimentally.

Reviewer #1 (Comments to the Authors (Required)):

In the manuscript by Grudtsyna, V. et al the authors apply computational and cellular biology approaches to assess the role of orientational order on cytoskeletal transmission of forces from ECM-engaged integrin adhesion receptors. This work builds upon a previous publication (Swaminathan, V. et al, 2017, PNAS) where the constrained aV integrin probe was applied to demonstrate that integrins are aligned by the forces applied by the actin cytoskeleton in conditioned with directional motion. The more recent findings in the current work support this alignment of integrins by the actin cytoskeleton, finding that increased ligand (FN) density occurs in conjunction with an increase in orientational order. Furthermore, assessment of small and larger adhesions, as well as the uncoupling of myosin II activity through inhibition with Blebbistatin, revealed the importance of myosin II activity in detection and response to ligand density.

Comments on main points:

1. The first main point of the paper is that orientational order in both aV and F-actin increases with ligand density, and that this is linked to integrin activation state. To support this finding,

a. Related to Fig. S1, can the authors comment on the increased spreading that is occurring independently of changes in focal FA area in Fig. S1c? Like in Fig. 1f, was there also a decrease/increase in FA number?

We thank the reviewer for drawing attention to the focal adhesion morphometry data we presented in the manuscript. The initially reported analysis using the focal adhesion analysis server (FAAS) presented a lot of small puncta (background staining) in the cell, which were identified as focal adhesions, leading to an overall high number of FAs/cell in all conditions. We have now improved this analysis to more precisely quantify the changes in FA number in the different conditions and set a standardized size threshold ($0.10\mu\text{m}^2 < \text{FA size} < 5\mu\text{m}^2$) for FAs for segmentation. This size threshold is consistent with the one we have used for segmenting FAs for analyzing our anisotropy data (using our own segmentation algorithms). To clarify this in the text, we have added these thresholds used for segmentation of FAs in the methods section.

"Paxillin images, taken as previously described in TIRF mode, were submitted for analysis to the online Focal Adhesion Analysis Server (FAAS)[1]. The adhesion size options were set to $0.10\mu\text{m}^2 < \text{FA} < 5\mu\text{m}^2$."

Based on these size thresholds, we don't find a significant increase in median FA area in cells plated at $0.1\mu\text{g/ml}$ FN and cells pretreated with Manganese on $0.1\mu\text{g/ml}$ FN (Figure S1b). However, we do find that increasing FN density or activating with Manganese on low FN density leads to a significant increase in the number of FAs per cell (Fig 1f, Fig S1b).

We have updated these results in the figures and added the following in the text:

"Consistent with previous results, we found that increase in FN density resulted in a gradual increase in cell spread area as well as in nuclear to cytoplasmic YAP ratio across all 3 densities (Figure. 1e). Locally, these FN density dependent changes coincided with small changes in median FA size and a significant increase in the overall number of FAs formed per cell(Figure. 1f). To test if these effects on cell and FA response was indeed a result of changes in FN density induced integrin activation, we measured the cell response of cells pretreated with Mn^{2+} plated on low FN density and found that similar to the increase in F-

actin orientational order, there was a significant increase in cell spread area compared to untreated cells on the same FN density as well as an increase in the number of FAs formed per cell (Figure. S1b)."

b. Further to Fig. S1, given the link between adhesion signalling and YAP nuclear localisation, can the authors confirm whether this increase in cell spreading from Mn²⁺ treatment also results in an increase in YAP nuclear localisation? Can the authors also provide representative images for the data in Fig. S1c to help the reader to better appreciate the significance of the differences?

We have now included representative images of MEFs pretreated with 1mM Mn²⁺ and then plated on 0.1µg/ml FN and immunostained for F-actin, nucleus and YAP (Fig S1c). Interestingly, while we find a robust increase in cell area when cells are pretreated with Mn²⁺(Fig S1c), this increase in cell area and changes in number of FAs/cell does not lead to a significant change in nuclear localization of YAP compared to untreated cells. This suggests that changes in integrin conformation induced by Manganese leads to increase in orientational ordering of integrins and F-actin and changes in cell spread area but independent of the integrin-Yap signaling pathway. We have clarified this in the text:

"However, pretreatment with Mn²⁺ on 0.1 µg/ml FN did not lead to a significant change in nuclear to cytoplasmic YAP ratio (Fig. S1c)."

2. In Fig. 2, the authors compare nascent and larger focal adhesions based on the relative areas. This data is quite difficult to evaluate, as the smallest bin for the 0.1 ug/ml FN has a very poor R² for the fit. Is it possible to increase the size of the bin to include the middle values, where the R² is consistently higher than 0.9, by increasing the size range from 0.1-1 um² to improve the fit quality for the 0.1 ug/ml.

We picked these 3 bin sizes based on previously published studies that have categorized focal adhesions based strictly on adhesion size into nascent adhesions (NAs), focal complexes (FCs) and focal and fibrillar adhesions[2], [3]. Regarding the relatively low R² value for the bin size 0.1<x<0.25 um² at 0.1ug/ml, both, the poorer fit as shown by the low R² value as well as the low amplitude (A) together signify a highly disorganized F-actin in these adhesions under this condition. It is even more striking if we consider that for the smallest bin, increasing the FN density (Fig2A, left column, top to bottom) increases the R² value and the amplitude(A), suggesting it is not specifically due to the bin size but due to FN density. Taken together, this suggests that the low R² value at 0.1ug/ml FN for adhesions in the size bin is a feature of organization. We have clarified this in the text:

"In fact, the modulation fit on the lowest FN density was very poor (R²=0.268) leading to an unreliable measure of orientational order. Since the quality of the fit depended only on the FN density and not on the actual bin size, this poor fit at the lowest density suggests that the smallest adhesions at this density have a very disordered F-actin organization (Figure. 2a)."

We include here for the reviewers, plots of F-actin orientational order where we combine the smallest and the middle-size bins and compare it to the large size adhesions (as suggested by the reviewer). The interpretation of this data remains the same i.e., "Orientational ordering of F-actin the small adhesions is the most sensitive to changes in ECM density". We are happy to include these in the supplementary figures if needed to further clarify our interpretation.

3. Also, it is unclear from the figure legends the number of separate biological replicates that were conducted. Could the authors add this detail into the legends to help the reader to better evaluate the data? This also brings with it the issue of cell numbers in many experiments being very low. I can understand that for the polarisation microscopy, the numbers are suitably high as individual FAs are being assessed, but for the YAP N/C ratio assessment, as well as the cell size and FA #/cell (Fig. 1e/f and Fig. 3b), the cell numbers are far too low to draw such strong conclusions. I would hope to see at least 60 cells (20 from each of three biological replicates) for each condition before publication.

We agree with the reviewer regarding the lack of clarity in the figure legends and have now rewritten them to make it clear. Each anisotropy curve in Figure 1 and Figure 3 represents one experiment and comparison of orientational order across experiments is in Figure S1 and Figure S2 for all the conditions. In addition, we have performed more experiments for YAP quantification from at least 3 biological repeats for all conditions in Figures 1 2 and 3 adding to the statistical robustness of these readouts.

Reviewer #2 (Comments to the Authors (Required)):

The work « Extracellular matrix sensing via modulation of orientational order of integrins and F-actin in focal adhesions » by V. Grudtsyna et al. uses a method already developed by the authors, based on polarized fluorescence imaging to explore the intrinsic relation between the orientational order of these proteins and the properties of the extra cellular matrix (ECM rigidity and density). The use of this method is made possible by the measurement of

integrin and actin proteins with appropriate labelling, e.g. which exhibit sufficiently reduced angular fluctuation, thanks to reduced linker length and flexibility between the fluorophore and protein of interest. Despite remaining fluctuations, the method allows in principle for quantitative results when comparing different situations involving the same protein construct. The authors perform several test experiments (including varying the properties of the ECM substrate and actomyosin contractility) and based on a model, conclude that the orientation of focal adhesion (FA) proteins participate in the regulation of mechanotransduction, which is the main result of the work. The work is of interest and the application of the method brings extra knowledge on the orientational organization of proteins, which is a need when addressing the main question of the relation between orientation and mechanotransduction. There are however points that need to be better addressed in order to support the conclusion of the work.

1. *The method used in the work has the advantage of being simple, however it presents several inconvenients that make its capabilities of quantification challenging. This is not addressed in the manuscript :*
 - a. *The method requires a lot of different orientation angles of FAs being measured to extract a reliable anisotropy (order) value from the measured proteins. What is the required number of angles measured to be able to accurately measure an anisotropy value ? finding FAs of different orientations can be difficult and require a lot of data to be recorded, how is the possible heterogeneity between FAs taken into account in the anisotropy curves results ?*

The reviewer is spot on that while a major advantage of this method is that it is simple and easy to interpret, the robustness of the anisotropy curve depends on the range of FA angles sampled during the measurement. This is similar to having different polarization angles of excitation light in traditional polarization microscopy, where increasing the number of polarization angles can increase the “angular” resolution of the measurements. The relationship between the robustness in the measure of orientational order and the number of FA angles depends on several factors including the anisotropy of the biological molecule/fluorescence probe being imaged and the *G-factor* of the optical system which considers the “cleanness” of polarization of excitation light and the efficiency of the detection system. In addition, when comparing across different conditions (such as ECM density here), the organizational sensitivity of the molecule to changing conditions is another factor. In our case, since F-actin is an inherently anisotropic polymer, the number of angles required for getting a robust orientational order of F-actin or changes in order as a function of anisotropy curve is much lesser than for integrins which is a dynamic membrane receptor with indirect linkage to the F-actin. Below, for the reviewer we include plots of F-actin anisotropy as a function of FA angle for 10ug/ml FN with 3 different angle bins (6, 12, 20).

From below, it is clear that even at the lowest number of FA bins (6), the difference in F-actin orientation as a function of fibronectin density is quite robust and significant.

No. of FA bins=6

We are happy to add this information in the supplementary figures if recommended so by the reviewers.

We also agree with the reviewer, that grouping all FAs in a cell in a single plot as we do does average out the information about the heterogeneity of FAs in a cell under a given condition. Figure 2 where we bin FAs based on size is one such effort to explore this heterogeneity and it led us to explore the role of myosin II in this study. It is highly likely that within each FA sub-population, depending on the specific composition or state of FA, this will also change. To highlight this, we have added this in our discussion section:

“This may correlate to conditions that require higher contractility such as the higher ECM density here or high stiffness ECMs or it may reflect a compositional switch in FAs driven by changes in integrin subtype from $\alpha V\beta 3$ integrin to $\alpha 5\beta 1$ integrin. The precise relationship between organizational (orientational order of FA components) and functional subpopulations of FAs (fibrillar vs focal adhesions or front vs rear) will be areas of future investigations.”

b. The experiment is an intrinsic projection of the information in 2D, since pure polarization projections are used. This means that if the disorder is low but is tilted in 3D, this will result in a measurement that shows a high disorder, due to the well problem issue of bias introduced by 3D orientation of the measured organizations. How do the authors ensure that this is not an issue? can some of the measurements, in particular decrease of anisotropy values, be affected by 3D tilt rather than disorder, in particular when measuring integrin (which 3D orientation varies a lot) and actin (which is tilted in the FA structure)?

This is an excellent point from the reviewer that we cannot extract any information about the tilt of the molecule and as the reviewer pointed out it is possible that under different conditions of ECM densities or contractility, F-actin and integrin tilt also changes. While we cannot determine the contribution of tilt, since our excitation is *s-polarized*, and all the conclusions we make come from the modulation in anisotropy relative to this plane of polarization (in-plane), we can conclude with confidence that the in-plane orientational order of F-actin and integrins does indeed vary with ECM density. Taken together with the reviewers' comments, this suggests important role for 2D orientational order and 3D axial tilt in mechanosensing, which will be important to understand in the future. Newer approaches using single molecule based polarization techniques or using linkers of different length can help in addressing this in the future. To clarify this in the text, we have added this to our discussion section:

“In addition to molecular anisotropy along the plane of the membrane, it is likely that owing to the 3D nature of the FA architecture, the orientational of molecules perpendicular to the membrane(tilt) may also play a significant role in ECM sensing. Our approach here with a single probe and measurement of 2D projection of the anisotropy is currently limited in extracting the contribution of tilt, though other approaches which rely on single molecule polarization or using linkers of different length can address this[4]–[6]. Consistent with this 3D organization hypothesis, our previous measurements on integrin anisotropy using 2 different polarization probes suggests an important role for tilting of integrins upon activation [7].”

c. The measured anisotropy value is a mixture of what is due to structural disorder and what is due to the orientational freedom fluctuation of the fluorescent molecules attached to the protein of interest. Is there a good proof that this fluctuation does not change in the different measurements performed by the authors ? could the anisotropy changes measured be due to a modification of this fluctuation behavior, due to forces exerted on proteins (which is necessarily happening in the studied process), or due to different packing properties ?

Changes in the rotational mobility of the fluorophore can change due to changes in molecular packing in a dense structure such as the focal adhesion or due to changes in the flexibility of the structure due to forces as mentioned by the reviewer. Previously, our group and others have reported good agreement between in-vitro polarization measurements and in-vivo (in cells) measurements of AF488-labelled F-actin. In addition, reports ([5], [8]) have recently shown that wobbling of the fluorescent dye in phalloidin is not sensitive to actin filament packing within bundles due to the small size of the AF488 molecule. For measurements of integrin, the fluorescent molecule on the integrin probe is on the extracellular domain of the molecule where packing/crowding is less of an issue. We previously showed no dependence of total fluorescence intensity to the alphaV-const. to its anisotropy values to rule out contribution of packing and homo-FRET in our measurements[7]. However, as the reviewer has pointed out, changes in structural flexibility due to forces can potentially alter anisotropy, and while experimental data on this is currently lacking, we cannot rule it out. To clarify this, we have added:

“ Our model here shows that when the ECM dependent modulation of orientational order measured is coupled to directional catch bond, this can lead to highly sensitive mechanosensing of ECM cues which we also experimentally capitulate. However, it is also likely that other mechanisms contribute to establishment and regulation of orientational order. For e.g., at the single molecule level, it is possible that ECM-dependent forces acting on integrins or F-actin can directly alter the structural flexibility of the molecules resulting in changes in its orientational order.”

2. The number of data point in modulation measurements is not clear: it is said that there are about 1000 FAs measured from collections of 35 to 80 cells (for instance in figure 1), however this would mean about 20 to almost 40 FAs per cells, which does not seem to be the case in some of the situations shown in Fig. 1a,c.

We apologize for the confusing legends and numbers stated for different figures. We have added more detailed information about each experiment and improved our segmentation of FAs with appropriate filtering (see response to point 3 below). In general, we find on an average approximately 25, 50 and 50 FAs/ cell as we go from 0.1ug/ml to 1 and then 10ug/ml FN (Fig 1f). With regards to the mismatch with images in Fig 1a or 1c, it is especially the case with 10ug/ml FN condition, that a complete cell does not fit into the field of view (due to an image splitter in EA-TIRFM). It is for this reason that FA morphometry is done in a non-EA-TIRFM system where one can capture the full field of view.

3. Fig. 1f shows in the label a number of FA per cell of 600, this is very surprising.

As mentioned in response to reviewer 1, The initially reported analysis using the focal adhesion analysis server (FAAS) presented a lot of small puncta (background staining) in the cell, which were identified as focal adhesions, leading to an overall high number of FAs/cell in all conditions. We have now improved this analysis to more precisely quantify the changes in FA number in the different conditions and set a standardized size threshold ($0.10\mu\text{m}^2 < \text{FA size} < 5\mu\text{m}^2$) for FAs for segmentation.

“Paxillin images, taken as previously described in TIRF mode, were submitted for analysis to the online Focal Adhesion Analysis Server (FAAS)[1]. The adhesion size options were set to $0.10\mu\text{m}^2 < \text{FA} < 5\mu\text{m}^2$ ”

4. The measurement of the orientation of the measured FAs is not clearly explained. This orientation seems to be reported with a high precision in the anisotropy curves, while there can be a strong error bar around a given FA orientation. In small FAs in particular the shape does not allow to visualize a clear orientation, its value can be defined with variations up to 40 degrees, how is the mean orientation error accounted for ?

We apologize for not clearly stating our selection criteria for segmented FAs in cell chosen for further analysis. For analyzing polarization images, we used our own image analysis tools for segmentation of the adhesion channel. To the segmented objects, we then applied a threshold for size ($0.10\mu\text{m}^2 < \text{FA size} < 5\mu\text{m}^2$) and eccentricity (< 0.9). The latter was applied for the exact reason the reviewer pointed out, which was to get reliable measure of orientation of the FA. We have clarified this analysis process in the methods section of the text now:

“FAs were segmented from the background subtracted paxillin image using the “Moments” binarization algorithm from the ImageBinarization.jl package. The FA mean anisotropy and the angle between the FA long axis and the excitation polarization (θ), were extracted using in-house packages in Julia. FAs between $0.1 - 5 \mu\text{m}^2$ and with eccentricity less than 0.9 (and less than 0.7 in the Bb washout experiments) were filtered out for reliable estimation of FA long axis orientation”

5. The type of actin population chosen for the measurement is not clearly defined with respect to the question addressed in this work. Shouldn't actin be investigated at the FA proximity ? in the images shown in Fig. 1 and Fig. 3, there seem to be a lot of actin populations from peripheral stress fibers, rather ventral stress fibers which involve more clearly the connection to FAs.

The goal of the present study was to investigate the nanoscale organizational link between integrins and F-actin within a FA and its role in ECM sensing. As pointed out by the reviewer, there also seems to be a link between changes in ECM and/or integrin activation and F-actin population around the cell and not just in the FA. We are in fact investigating this relationship by specifically looking at F-actin architecture linking the FA to the nucleus. We have added the following in the discussion:

“Our EA-TIRFM images also reveals a strong correlation in the overall architecture of F-actin in the cell, away from the FAs, and the ECM dependent changes in orientational order of F-actin in FAs. This suggests an organizational link between nanoscale orientational ordering of FA components and mesoscale F-actin organization. Interestingly, long range mesoscale order of actin in cells has been previously shown to play a significant role in modulating ECM

dependent changes in the cell's mechanical properties[55]., which mechanistically could be linked to orientational order of F-actin in FAs.”

6. In the clutch model developed in the Supplementary Note on computational model, what seems to be directional is the direction of actin flow, while in the rest of the paper molecular orientation is driven by the probability P_v introduced in the model. The authors should make clearer where the orientation of actin vs. flow comes into play in the model. It is also not clear how the integrin directionality measured in the experimental section is related to directionality in the theoretical model.

We apologize to the reviewers for this confusion and the lack of clarity in our description of the directionality in the model. In our model, ECM-dependent orientational order of actin is incorporated by changing the pathway for vinculin unbinding actin, which determines the lifetime of the clutch/ligand bond (figure 4b). The probability by which vinculin unbinds actin follows either the directional (high lifetime) or non-directional (low lifetime) pathway. Previously, it is this directional interaction that has been shown to lead to long-range order of actin in migrating cells. Therefore, when orientational order of actin is high, the directional pathway for vinculin unbinding is used in the model. By contrast, when orientational order of actin is low, the non-directional pathway for vinculin unbinding is used. If ECM density increases the orientational order of actin, then the probability of directional pathway also increases with ECM density in the model. In other words, the model incorporates our experimental measurement of orientational order of F-actin changes in an ECM dependent manner by changing the probability of vinculin unbinding actin, P_v . The actin flow direction is not varied and is used exclusively to build tension on the vinculin-F-actin bond.

To make these details clear, we have added a section entitled “**Directional vinculin catch bond kinetics indicates actin orientation**”, which explicitly focuses on how actin flow versus orientation affects clutch dynamics. This section is included in the Supplementary Note on computational model.

A few explanatory sentences are also incorporated in the main text on lines ...:

“Two implicit components were also included in each clutch: actin with a given orientation and retrograde flow velocity, and vinculin, with a specific pathway for unbinding actin, which depended on actin orientation.”

7. The finding that 'Orientational order of FA components increases sensitivity for ECM ligand binding by modulating directional catch bonds downstream of integrins.' Is the result of simulations, and it is not clear how this model is related to the data measured in the rest of the work. This is an important conclusion since the main point of the manuscript is to address if FA molecules orientation is the result or the cause of the mechanosensitive process leading to integrin activation.

Orientational ordering of FA molecules is both the result and the cause of ECM ligand sensitivity. Integrin is sensitive to ECM density by increasing its activation, but this mechanism is not sufficient for explaining the sensitivity of whole adhesions to ligand density. Actin orientational order is also sensitive to ECM density, but, again, this mechanism alone cannot explain ligand sensitivity. The model tests the hypothesis that these known mechanisms that, together drive orientational order of FA components (activation for integrins and directional catch bonds for F-actin) result in increased sensitivity of adhesions to ECM density. We demonstrate that increasing both integrin activation and orientation-dependent actin unbinding of adhesion molecules regulates the sensitivity of adhesions to ECM density. This mechanism is uncoupled to stiffness sensing. To summarize, our results demonstrate ECM-dependent integrin activation or orientation-dependent actin unbinding alone do not

regulate adhesion sensitivity to ECM density. By contrast, when combined, ECM-dependent integrin activation and directional actin unbinding from adhesion molecules regulate the sensitivity of adhesions to ECM density. The model is related to the data measured in the rest of the work because it first includes both changes in integrin activation with ECM density in orientational order of F-actin that were observed experimentally. In particular, as shown by the Manganese experiment as well as in our previous publication using ECM proteins vs. PLL and mutations in talin ([7]), changes in orientation order of integrin is incorporated as changes in activation rate, P_a , which increases linearly with ligand concentration. Because orientational order of F-actin is established in FAs downstream of integrin activation, the orientational order of actin is incorporated using a direction-dependent catch bond kinetics for vinculin unbinding actin in the adhesions. Second, our model demonstrates that orientational order of integrin through ECM-dependent activation and orientational order of actin via vinculin unbinding, alone are not sufficient to determine ligand sensitivity, but if combined, they determine ligand sensitivity. Therefore, result from the model, supported by the experiments is that both the ECM-dependent integrin activation and ECM-dependent actin unbinding from adhesions confer sensitivity of adhesions to ECM density. Third, the model is used to demonstrate that the sensitivity of the adhesions to ECM densities is uncoupled from their sensitivity to substrate stiffness. These predictions from the model are tested experimentally in the last part of the paper.

Similar to our initial write-up on orientational order of F-actin in the model, we did not write about the coupling between integrin orientational order, integrin activation and the model in the main text. We have now added another section in the supplementary note for this. Again, our central conclusion here is that known mechanisms that drive orientational order of FA components (activation for integrins and directional catch bonds for F-actin) result in increased sensitivity to ECM sensing. Taken separately, these mechanisms do not increase ligand sensitivity.

Integrin activation rate indicates integrin directionality

Integrins switch between inactive and active states, and only when active they form a bond with free substrate ligands. Our current and previous experimental measurement directly links ECM-dependent integrin activation with orientational order of integrins in FAs. The model therefore couples the probability of activation, P_a , with the change in orientational order measured in the experiments.

Directional vinculin catch bond kinetics indicates actin orientation

In the model, actin filament, actin flow and vinculin proteins are modeled implicitly through v_{flow} , and k_{off} . The flow velocity, v_{flow} , displaces ligated clutches to build tension on their bonds with substrate ligands. Depending on the magnitude of this tension, a number of vinculin is considered (2 for forces below 8pN; 5 for forces between 8 and 15 pN; 9 for forces between 15 and 21 pN; 11 for forces > 21 pN). The pathway for vinculin unbinding (which determines the unbinding rates given certain force values) is chosen, in each run, depending on the probability of directionality that the model is testing. When it is assumed that actin filaments are directional (barbed end pulled), the first pathway (with maximum bond lifetime equal to 13 s) is used. Viceversa, when actin filaments are not directional (pointed end pulled), the second unbinding pathway (with maximum bond lifetime equal to 3 s) is used for all vinculins. For intermediate probabilities of directional versus non directional actin filaments, then a proportional number of vinculins will unbind with one pathway and the remaining will follow the second pathway.”

8. The authors could probably consider in the discussion part of their work, how the question addressed in the previous point can be addressed experimentally.

The strength of the model is that it uses the readout of orientational order, couples it to known mechanisms that are known to drive orientational order of F-actin and integrins and clearly shows an improvement in ECM sensitivity (which we experimentally validate). We also agree with the reviewer that the mechanisms used here is one of potentially several such mechanisms including possibility of orientational organization of formins, localization of actin cross-linking proteins and other directional catch bonds. We have added these points in our discussion section including specific experiments that can address these questions:

” Currently, the specific mechanism that results in F-actin orientational order downstream of integrin activation and integrin orientational order is unknown. In addition, while the link between ECM-dependent integrin activation and its orientational order is robust, we still don’t have a direct method to alter orientational order of integrins independent of activation. With advancements in ECM nano-deposition and nanofabrication techniques, this challenge can potentially be overcome in the future. To address the question of the link between F-actin and integrin orientational order, a computational model for directionally asymmetric catch bonds between vinculin and F-actin has previously shown the establishment long range F-actin orientational order[10]. In our study, we show that the modulation of orientational order measured here when linked to that directional catch bond as previously shown can lead to highly sensitive mechanosensing of ECM cues. However, it is also likely that other mechanisms contribute to establishment and regulation of orientational order. For e.g., at the single molecule level, it is possible that ECM-dependent forces acting on integrins or F-actin could be directly altering the structural flexibility of the molecules resulting in changes in orientational order. At the multimolecular scale, ECM dependent regulation of other directional catch bonds such as between talin and F-actin or localization of actin crosslinking proteins which have been shown to promote F-actin orientational order in focal adhesions could be driving this process[11]–[13]. Computation modeling coupled with point-mutagenesis studies in our group are currently aimed at testing the role of these specific interactions. Lastly, mechanistically it is also tempting to speculate that orientational ordering of actin nucleators such as FA-specific formins can lead to F-actin orientational order in an ECM dependent manner, though this requires a detailed studies on different FA specific formins. Interestingly, long range order of actin in cells has been previously shown to play a significant role in modulating ECM dependent changes in its mechanical properties[55] and our imaging also reveals a correlation in the overall architecture of F-actin in the cell and the ECM dependent changes in orientational order of F-actin in FAs. This link between nanoscale orientational ordering of FA components and mesoscale F-actin organization and its functional relationship with mechanical properties cells remains to be investigated.”

- [1] M. E. Berginski, S. M. Gomez, J. Jones, and K. Yamada, “The Focal Adhesion Analysis Server: a web tool for analyzing focal adhesion dynamics,” *F1000Research* 2013 2:68, vol. 2, p. 68, Mar. 2013, doi: 10.12688/f1000research.2-68.v1.
- [2] C. K. Choi, M. Vicente-Manzanares, J. Zareno, L. A. Whitmore, A. Mogilner, and A. R. Horwitz, “Actin and α -actinin orchestrate the assembly and maturation of nascent adhesions in a myosin II motor-independent manner,” *Nat. Cell Biol.*, vol. 10, no. 9, pp. 1039–1050, 2008, doi: 10.1038/ncb1763.
- [3] M. L. Gardel, I. C. Schneider, Y. Aratyn-Schaus, and C. M. Waterman, “Mechanical integration of actin and adhesion dynamics in cell migration.,” *Annu Rev Cell Dev Biol*, vol. 26, pp. 315–333, 2010, doi: 10.1146/annurev.cellbio.011209.122036.
- [4] T. Ding and M. D. Lew, “Single-Molecule Localization Microscopy of 3D Orientation and Anisotropic Wobble Using a Polarized Vortex Point Spread Function,” *Journal of Physical Chemistry B*, vol. 125, no. 46, pp. 12718–12729, Nov. 2021, doi: 10.1021/ACS.JPCB.1C08073/ASSET/IMAGES/LARGE/JP1C08073_0005.JPEG.
- [5] C. V. Rimoli, C. A. Valades-Cruz, V. Curcio, M. Mavrakakis, and S. Brasselet, “4polar-STORM polarized super-resolution imaging of actin filament organization in cells,”

- Nature Communications* 2022 13:1, vol. 13, no. 1, pp. 1–13, Jan. 2022, doi: 10.1038/s41467-022-27966-w.
- [6] M. Kampmann, C. E. Atkinson, A. L. Mattheyses, and S. M. Simon, “Mapping the orientation of nuclear pore proteins in living cells with polarized fluorescence microscopy,” *Nat Struct Mol Biol*, vol. 18, no. 6, pp. 643–649, 2011, doi: 10.1038/nsmb.2056.
- [7] V. Swaminathan *et al.*, “Actin retrograde flow actively aligns and orients ligand-engaged integrins in focal adhesions.,” 2017. doi: 10.1073/pnas.1701136114.
- [8] S. B. Mehta *et al.*, “Dissection of molecular assembly dynamics by tracking orientation and position of single molecules in live cells.,” *Proceedings of the National Academy of Sciences*, pp. 1–25, 2016, doi: 10.1101/068767.
- [9] M. Gupta *et al.*, “Adaptive rheology and ordering of cell cytoskeleton govern matrix rigidity sensing,” *Nat Commun*, vol. 6, no. 1, p. 7525, Dec. 2015, doi: 10.1038/ncomms8525.
- [10] D. L. Huang, N. A. Bax, C. D. Buckley, W. I. Weis, and A. R. Dunn, “Vinculin forms a directionally asymmetric catch bond with F-actin,” *Science (1979)*, vol. 357, no. 6352, 2017.
- [11] L. M. Owen, N. A. Bax, W. I. Weis, and A. R. Dunn, “The C-terminal actin-binding domain of talin forms an asymmetric catch bond with F-actin,” *Proc Natl Acad Sci U S A*, vol. 119, no. 10, Mar. 2022, doi: 10.1073/PNAS.2109329119.
- [12] M. A. Juanes, D. Isnardon, A. Badache, S. Brasselet, M. Mavrikis, and B. L. Goode, “The role of APC-mediated actin assembly in microtubule capture and focal adhesion turnover,” *Journal of Cell Biology*, vol. 218, no. 10, pp. 3415–3435, Oct. 2019, doi: 10.1083/JCB.201904165/VIDEO-4.
- [13] O. Loison *et al.*, “Polarization-resolved microscopy reveals a muscle myosin motor-independent mechanism of molecular actin ordering during sarcomere maturation,” *PLoS Biol*, vol. 16, no. 4, p. e2004718, Apr. 2018, doi: 10.1371/JOURNAL.PBIO.2004718.

June 21, 2023

RE: Life Science Alliance Manuscript #LSA-2023-01898-TR

Dr. Vinay Swaminathan
Lund University
BMC D14
Sölvegatan 19
Lund, Skåne 223 62
Sweden

Dear Dr. Swaminathan,

Thank you for submitting your revised manuscript entitled "Extracellular matrix sensing via changes in orientational order of integrins and actin in adhesions". We would be happy to publish your paper in Life Science Alliance pending final revisions necessary to meet our formatting guidelines.

- Please include a conflict of interest statement in your main manuscript text.
- Please confirm that Figure S2 only has one panel. If so, labeling it as A is unnecessary.
- Please add your supplementary figure legends to the main manuscript text after the legends for the main figures.
- The manuscript has callouts for Figure S6A-C but there is no Figure S6. Please correct.
- Please add a callout for Figure S3 to your main manuscript text.
- please incorporate the Supplemental Methods and References into the main Methods and References sections

Figure checks:

- please add scale bars to Figure 1A and Figure S1 panels A&B

A. FINAL FILES:

B. MANUSCRIPT ORGANIZATION AND FORMATTING:

Sincerely,

Reviewer #1 (Comments to the Authors (Required)):

The authors have addressed my concerns

July 3, 2023

RE: Life Science Alliance Manuscript #LSA-2023-01898-TRR

Dr. Vinay Swaminathan
Lund University
BMC D14
Sölvegatan 19
Lund, Skåne 223 62
Sweden

Dear Dr. Swaminathan,

Thank you for submitting your Research Article entitled "Extracellular matrix sensing via changes in orientational order of integrins and actin in adhesions". It is a pleasure to let you know that your manuscript is now accepted for publication in Life Science Alliance. Congratulations on this interesting work.

DISTRIBUTION OF MATERIALS:

Again, congratulations on a very nice paper. I hope you found the review process to be constructive and are pleased with how the manuscript was handled editorially. We look forward to future exciting submissions from your lab.

Sincerely,
